



# Spatial and seasonal variability in volatile organic sulfur
# compounds in seawater and overlying atmosphere of the Bohai
# and Yellow Seas
**Juan Yu[1,2,3,†], Lei Yu[1,†], Zhen He[1,2,3], Gui-Peng Yang[1,2,3,*], Jing-Guang Lai[1], Qian Liu[1]**
[1]Frontiers Science Center for Deep Ocean Multispheres and Earth System, Key Laboratory of Marine Chemistry
Theory and Technology, Ministry of Education, Ocean University of China, Qingdao 266100, China.
[2]Laboratory for Marine Ecology and Environmental Science, Qingdao National Laboratory for Marine Science and
Technology, Qingdao 266237, China.
[3]Institute of Marine Chemistry, Ocean University of China, Qingdao 266100, China.
**Abstract.** To better understand the production and loss processes of volatile organic sulfur compounds (VSCs) and
their influence factors, VSCs including carbon disulfide ($CS_2$), dimethyl sulfide (DMS), and carbonyl sulfide (COS)
were surveyed in the seawater and atmosphere of the Bohai and Yellow Seas during spring and summer of 2018. The
concentration ranges of COS, DMS, and $CS_2$ in the surface seawater during spring were 0.14–0.42, 0.41–7.74, and
0.01–0.18 nmol $L^{-1}$, respectively, and 0.32–0.61, 1.31–18.12, and 0.01–0.65 nmol $L^{-1}$ during summer. COS and $CS_2$
had high concentrations in coastal waters, which may be due to elevated photochemical production rates. High DMS
concentrations occurred near the Yellow River, Laizhou Bay, and Yangtze River Estuary coinciding with high nitrate
and Chl *a* concentrations due to river discharge during summer. The depth distributions of COS, DMS, and $CS_2$ were
characterized by high concentrations in the surface seawater that decreased with depth. The mixing ratios of COS,
DMS, and $CS_2$ in the atmosphere were 255.9–620.2 pptv, 1.3–191.2 pptv, and 5.2–698.8 pptv during spring, and

* Corresponding author at: Key Laboratory of Marine Chemistry Theory and Technology, Ministry of Education, Ocean University of China, 238 Songling Road, Qingdao 266100, China. E-mail address: gpyang@mail.ouc.edu.cn (G.-P. Yang)

† These authors contributed equally to this work and should be considered co-first authors.





394.6–850.1 pptv, 10.3–464.3 pptv, and 15.3–672.7 pptv in summer. The mean oceanic/atmospheric concentrations
of COS, DMS, and $CS_2$ were 1.8/1.7-, 3.1/4.7-, and 3.7/1.6-fold higher in summer than spring due to the high Chl *a*
concentrations in summer. The mean sea-to-air fluxes of COS, DMS, and $CS_2$ were 1.3-, 2.1-, and 3.0-fold higher in
summer than spring. The sea-to-air fluxes of VSCs indicated that these marginal seas are major sources of VSCs in
the atmosphere. The results provide help with a better understanding of the control of VSCs distributions in marginal
seas.
**Keywords:** Volatile organic sulfur compound; Carbonyl sulfide; Dimethyl sulfide; Carbon disulfide



## 1 Introduction

Carbonyl sulfide (COS), dimethyl sulfide (DMS), and carbon disulfide (CS$_2$) are three major volatile organic sulfur compounds (VSCs) in seawater and the marine atmosphere and their biogeochemical cycles are closely related to climate change (Charlson et al., 1987; Li et al., 2022). VSCs are important in the formation of atmospheric cloud condensation nuclei (CCN) and sulfate aerosols, which have significant implications in the global radiation budget and ozone concentration (Andreae and Crutzen, 1997). COS can be converted to sulfate aerosols in the stratosphere and affect Earth's radiation balance (Crutzen, 1976). Atmospheric DMS can react with OH and NO$_3$ radicals to form SO$_2$ and methane sulfonic acid (MSA, CH$_3$SO$_3$H), and then form non-sea salt sulfates (nss-SO$_4^{2-}$), all of which contribute to acid deposition and CCN (Charlson et al., 1987). CS$_2$ is the key precursor of COS and 82% COS is the oxidation production of CS$_2$ (Lennartz et al., 2020). Hence, interest in the distribution, production, and chemistry of VSCs has grown in recent years (Lennartz et al., 2017; Lennartz et al., 2020; Li et al., 2022; Remaud et al., 2022; Whelan et al., 2018; Yang et al., 2008; Yu et al., 2022). Some researches indicates that the ocean is the source of VSCs (Chin and Davis, 1993; Yu et al., 2022). Opposite results also were reported that the ocean is the sink of VSCs (Zhu et al., 2019).

COS, with an average tropospheric residence time of 2–7 years, is the most abundant and widely distributed reduced sulfur trace gas in the atmosphere (Brühl et al., 2012). Atmospheric COS is mainly directly originated from oceanic emissions and indirectly from oxidations of dimethylsulfide (DMS) and carbon disulfide (CS$_2$) (Kettle et al., 2002; Lennartz et al., 2020). Uptake by terrestrial vegetation and soil is the most important sink of atmospheric COS (Kettle et al., 2002; Maignan et al., 2021). Therefore, COS is recognized as a proxy for estimating the photosynthesis rate in ecosystems (Campbell et al., 2008). DMS is the predominant biogenic sulfur originated from dimethylsulfoniopropionate (DMSP) which is mainly produced by bacteria and phytoplankton (Curson et al., 2017; Keller et al., 1989). Community composition of phytoplankton and bacteria can affect the net DMSP concentrations via synthesis and degradation (O'Brien et al., 2022, Zhao et al., 2021). DMS entering the atmosphere via sea-to-air exchange accounts for about 50% of all natural sulfur release (Cline and Bates, 1983). Photochemical reaction with dissolved organic matter is a principal source of CS$_2$ in seawater (Xie et al., 1998). The anthropogenic CS$_2$ source were rayon and/or aluminum production, fuel combustion, oil refineries, and coal combustion (Campbell et al., 2015; Zumkehr et al., 2018). Two different approaches (ice core (Aydin et al., 2020) and isotope measurements (Hattori et



al., 2020)) were used to evaluate anthropogenic COS emissions. The latter study and a modeling approach used by

Remaud et al. (2022) all found the gradient of anthropogenic COS in East Asia. Anthropogenic COS is initially emitted

as $CS_2$ and oxidized by OH to COS in the atmosphere (Kettle et al., 2002). The production and loss of VSCs involves

in phytoplankton and bacteria synthesis, zooplankton grazing, bacterial degradation, sea-air diffusion, photo-oxidation

and/or photochemical reaction (Schäfer et al., 2010). The mechanisms of production and loss processes still need to

be declared.

Yu et al. (2022) investigated the distributions of COS, DMS, and $CS_2$ and sea-to-air flux in Changjiang Estuary and

its adjacent East China Sea and showed that the oceanic VSCs (COS, DMS, and $CS_2$) are sources of atmospheric

VSCs. Different from Yu et al. (2022), Zhu et al. (2019) showed that the ocean is a sink for COS. The YS and BS are

semi-enclosed seas of the northwestern Pacific Ocean, the hydrological characteristics of this area are greatly affected

the BS coastal current, YS coastal current, and YS warm current (Chen, 2009), which may alter the VSCs distributions

via exchanges of water mass. In addition, the Yellow Sea Cold Water Mass (YSCWM), a seasonal hydrological

phenomenon, located in the 35 °N transect forms, peaks, and disappears in spring, summer, and after September,

respectively (Zhang et al., 2014). In this study, we investigate the spatial distributions and seasonal variability of COS,

DMS, and $CS_2$ in the seawater and overlying atmosphere of the Yellow Sea (YS) and Bohai Sea (BS) and the effects

of YSCWM on VSCs distributions to better understand the production and loss processes of VSCs.

**2 Materials and methods**

**2.1 Sampling**

Two cruises were conducted aboard the R/V "Dong Fang Hong 2" in the YS and BS from 28 March to 16 April

(spring) 2018 and from 24 July to 8 August (summer) 2018. The sampling stations are shown in Fig. 1. Seawater

samples were collected using 12 L Niskin bottles mounted on a Seabird 911 CTD (conductivity-temperature-depth)

rosette. The seawater was slowly siphoned from the Niskin bottles into 100 mL glass jaw bottles (CNW Technologies

GmbH, GER) via a translucent silicone tube. The seawater was allowed to overflow the sampling bottle by twice its

volume before the silicone tube was gently removed and the bottle was immediately sealed with an aluminum cap

containing a Teflon-lined butyl rubber septum without any headspace. After seawater the samples were collected, the

concentrations of oceanic VSCs were immediately measured on the ship. The environmental and hydrological



parameters such as seawater temperature and salinity were measured simultaneously by the CTD equipment.
Atmospheric samples were collected using cleaned and vacuumed SilcoCan canisters (Restek, USA) placed where
atmospheric VSCs were collected in the windward direction approximately 10 m above the ocean. The stability of
VSCs in this kind of fused silica-lined canister has been verified by previous studies (Brown et al., 2015). Brown et
al. (2015) verified the stability of VSCs during storage for 16 d at room temperature. The atmospheric samples were
analyzed immediately after being brought back to the laboratory.
**2.2 Analytical procedures**
VSCs in seawater were measured using a gas chromatographer (GC) (Agilent 7890A, USA) with a flame photometric
detector (FPD) and atmospheric VSCs were measured using a GC equipped with mass spectrometric detector (GC-
MSD) (Agilent 7890A/5975C, USA) according to the methods of Inomata et al. (2006) and Staubes and Georgii
(1993), respectively. A CP-Sil 5 CB column (30 m $\times$ 0.32 mm $\times$ 4.0 μm, Agilent Technologies, USA) was used to
separate the three VSCs. A standard curve was performed by measuring different concentrations of standard VSC
gases (COS, DMS, and $CS_2$ mixing ratios were 1 pptv, Beijing Minnick Analytical Instrument Equipment Center).
Qualitative analysis was conducted via comparison with the retention times of the standards, while quantitative
analysis was conducted by diluting the VSC standard gases to specific concentrations using the 2202A dynamic
dilution meter (Nutech, USA) and then injecting different volumes of the diluted VSC standards into the GC using a
gas-tight syringe. The VSC standard gases were used to provide a range of different concentrations and peak areas.
Based on the similarities between the concentrations and the peak areas of the standards and samples, the
concentrations of VSCs in the samples were obtained.
The concentrations of VSCs in seawater were determined using a cryogenic purge-and-trap system coupled with
the GC-FPD. A 30 mL seawater sample was injected into glass bubbling chamber by a gas tight syringe (SGE,
Australia). The VSCs were stripped from the seawater with high purity $N_2$ at a rate of 60 mL min$^{-1}$ for 15 min and
passed through an anhydrous $CaCl_2$-filled drying tube and a 100% degreasing cotton-filled 1/4 Teflon tube to remove
water and oxides. After that, the VSCs gases were passed through a six-way valve and trapped in a loop of a 1/16
Teflon capture tube immersed in liquid nitrogen. When the VSCs were all purged from the seawater, the capture tube
was removed from the liquid nitrogen and placed it into hot water (> 90 ℃) to desorb the trapped VSCs. At the same



time, VSCs gases were carried into the GC by $N_2$ and then detected by the FPD. The column temperature was
programmed with an initial temperature of 55 °C, followed by an increase to 100 °C at 10 °C min$^{-1}$, and a final increase
to 150 °C at 15 °C min$^{-1}$. The inlet and detector temperatures were 150 °C and 160 °C, respectively, and the split ratio
of pure $N_2$ was 10:1. The detection limit of the method for VSCs was 2.5~3.5 ng and the measurement precision was
3.2%~5.1% (Zhu et al., 2017).
The concentrations of atmospheric VSCs were analyzed using an Entech 7100 pre-concentrator (Nutech, USA)
coupled with the GC-MSD. A sampled SilcoCan canister was connected to the pre-concentrator and 200 mL of gas
was drawn into the preconcentration system with a three-stage cold trap. The pre-concentrator parameters of three-
stage cold trap are shown in Table S1. The first trap removes the $N_2$, $O_2$, and $H_2O$ (g) from the atmospheric samples
and the second trap eliminates the $CO_2$. As for the third trap, it is necessary to obtain the better peak shapes and
separation of the three VSCs. The temperature programming of the column was the same as for the seawater samples.
In addition, the temperature of the quadrupole and ion source were 110 °C and 230 °C, respectively, and the electron
ionization source was run at 70 ev. The carrier gas had a split ratio of 10:1 and a flow rate of 2.0 mL min$^{-1}$. Qualitative
and quantitative analysis of the VSCs was tested using the full scan mode (SCAN) and the selected ion monitoring
mode (SIM). The detection limit of the method for VSCs was 0.1~0.5 pptv (Zhu et al., 2017).
**2.3 Calculation of VSCs sea-to-air flux**
To estimate the VSCs volatilization from ocean to atmosphere, the sea-to-air fluxes of the VSCs were calculated
according to the model established by Liss and Slater (1974) with the following equation: $F = k_w(c_w - c_g/H)$, where $F$ is
the sea-to-air flux of VSCs (μmol m$^{-2}$ d$^{-1}$); $k_w$ is the VSCs transfer velocity (m d$^{-1}$); $c_w$ and $c_g$ are the equilibrium
concentrations of VSCs in the surface seawater and the atmosphere (nmol L$^{-1}$), respectively; and $H$ is the Henry's
constant calculated using the formula found in Table S2, which is converted to a dimensionless constant using the
conversion formula constructed by Sander (2015). $k_w$ was calculated by the N2000 method (Nightingale et al., 2000),
which has been internationally accepted, and the specific calculation introduced by Kettle et al. (2001).
**2.4 Nitrate, Chl a and dissolved organic carbon (DOC) measurement**
Seawater were filtered through Whatman GF/F filters (0.7 μm), and then the filtered water samples were stored at
−20 °C before nitrate analysis. A continuous flow analyzer (Skalar Analytical, Breda, Netherlands) were used to



measure nitrate concentrations. Chl *a* were extracted with 90% acetone for 24 h at 4 ℃ in darkness. Chl *a*
concentrations were determined following the method of Parsons et al. (1984) with a fluorescence
spectrophotometer (F-4500, Hitachi). Nitrate and Chl *a* data were provided by open research cruise supported by
NSFC Shiptime Sharing Project.
DOC concentrations were measured according to the method of Chen et al. (2021). Seawater were filtered through
Whatman GF/F filters (precombusted at 500 ℃ for 4 h), and the filtrate was stored at −20 ℃ for DOC analyses. The
DOC concentrations were determined by a total organic carbon analyzer (Shimadzu TOC-VCPH) after adding two
drops of 12 mol/L HCl in the DOC samples.
**2.5 Data analysis**
To better understand the sources and sinks of the three VSCs and the factors affecting their distributions, SPSS 24.0
software (SPSS Inc., Chicago, IL, USA) was used to analyze the relationships between environmental factors and the
three VSCs in seawater and atmosphere during spring and summer.
**3 Results**
**3.1 Spatial distributions of COS, DMS, and CS$_2$ in surface seawater**
**3.1.1 Spring distributions**
The temperature in the surface seawater showed a decreasing trend from south to north and the salinity increased from
the inshore to offshore sites due to the influences of the Yellow Sea warm current, Yalu River, and Yellow River (Fig.
2). The Chl *a* concentrations in the surface water of the BS and YS in the spring ranged between 0.17–4.45 μg L$^{-1}$
with an average of 1.19 ± 0.96 μg L$^{-1}$. The highest Chl *a* concentration (4.45 μg L$^{-1}$) occurred at station B39 in the
Bohai (Fig. 2), which may be related to the enhanced phytoplankton growth from the abundance of nutrients due to
the seawater exchange between the BS and YS. In addition, high Chl *a* concentrations were observed in the central
area of the southern YS.
The concentrations of COS, DMS, and CS$_2$ in the surface seawater of the BS and YS during spring ranged between
0.14–0.42, 0.41–7.74, and 0.01–0.18 nmol L$^{-1}$, respectively, with mean values of 0.24 ±0.06, 1.74 ±1.61, and 0.07 ±



0.05 nmol L$^{-1}$ (Fig. 2). The high COS concentrations observed during the spring all occurred in the YS (Fig. 2). The
highest COS concentration appeared at station H21 and coincided with a high Chl $a$ concentration. At the same time,
the two areas with high concentrations of COS in the central waters of the southern YS partially coincided with areas
with high Chl $a$ concentrations. High DMS concentrations existed in the coastal waters of the southern Shandong
Peninsula as well as at station B21 in the central part of the northern YS. The distribution of $CS_2$ in seawater exhibited
a decreasing trend from inshore to offshore (Fig. 2), which was similar with that of DOC. High $CS_2$ concentrations
appeared at stations H18 and H19 in the coastal waters of YSCWM (Fig. 2). There was also a high $CS_2$ concentration
at station B30 near the shore of the Liaodong Peninsula (Fig. 2).
**3.1.2 Summer distributions**
The temperature and salinity in the BS and YS in summer were relatively high and the high Chl $a$ concentrations were
concentrated in coastal waters (Fig. 3). The Chl $a$ concentrations in the seawater during summer ranged between 0.10–
4.74 µg L$^{-1}$ with an average of 1.60 ± 1.19 µg L$^{-1}$. Station B43 near the Yellow River estuary had a high Chl $a$
concentration (Fig. 3), which may have been due to the abundance of nutrients carried by nearby rivers or coastal
currents. There were low salinities, high nitrate and Chl $a$ concentrations at Stations H32, H34, and H35 in the
northeast of Yangtze River Estuary, and Stations B66 and B68 near the Laizhou Bay and Yellow River Estuary (Fig.
3 and S1).
The concentrations of COS, DMS, and $CS_2$ in the surface water of the BS and YS during summer were 0.32–0.61,
1.31–18.12, and 0.01–0.65 nmol L$^{-1}$, respectively, with mean values of 0.44 ± 0.06, 5.43 ± 3.60, and 0.26 ± 0.15 nmol
L$^{-1}$ (Fig. 3). The mean concentrations of Chl $a$, COS, DMS, and $CS_2$ were 1.3-, 1.8-, 3.1-, and 3.7-fold higher in
summer than spring. High COS concentrations were observed at stations B38 and B54 in the BS during summer. In
addition, COS had a high concentration at station H25 in the central part of the southern YS, which was near where
$CS_2$ also exhibited a high concentration (Fig. 3). High DMS concentrations were common in the northern BS, and
were generally coincident with high Chl $a$ levels. However, high Chl $a$ and DMS concentrations were found in the
coastal waters of the Yangtze River Estuary due to the Changjiang Diluted Water. In addition, the DMS concentration
was high at station H12 (Fig. 3). There were high $CS_2$ concentrations in the northeastern area of the Yangtze River
estuary (Fig. 3).





### 3.2 Depth distributions of COS, DMS, and CS$_2$ in seawater

#### 3.2.1 Depth distributions in spring

There was a clear tendency for temperature and Chl $a$ to gradually decreased from the surface to the bottom seawater (Fig. 4). The mean concentrations of Chl $a$, COS, DMS, and CS$_2$ were 5.4-, 5.1-, 5.9-, and 8.9-fold higher in the surface at about 4 m compared to > 60 m (Fig. 4). Consistent with the Chl $a$ distribution, the depth distribution of DMS in the seawater decreased from the euphotic zone to the bottom seawater (Fig. 4). The high COS concentrations occurred in the surface seawater and decreased with depth, and the lowest concentrations occurred in the bottom waters. CS$_2$ exhibited obvious depth gradients at most stations during spring, with higher concentrations in the surface except for station H15, where the CS$_2$ concentrations were high in the bottom seawater.

#### 3.2.2 Depth distributions in summer

In the 35 °N section there was clear influence of the YSCWM in summer. There were obvious temperature differences between the surface seawater and bottom seawater in summer, and stratification in the water bodies was observed (Fig. 5). A distinct thermocline emerged at a depth of 20 m, indicating that the YSCWM had formed (Fig. 5). The high Chl $a$ concentrations in the surveyed area of the BS and YS during summer all occurred in the euphotic zone, and the highest concentrations occurred in the waters at depths of 10–20 m (Fig. 5). The mean Chl $a$ concentrations were 5.4-fold higher at depths of 10–20 m compared to > 60 m. The depth distribution of DMS in seawater during summer decreased from the surface to the bottom seawater (Fig. 5). COS and CS$_2$ showed a significant depth gradient at most stations and decreased with the increasing depth. The mean concentrations of COS, DMS, and CS$_2$ were 17.5-, 21.6-, and 21.0-fold higher in the surface at about 3 m compared to > 60 m. However, in the bottom waters of station H16, COS had a relatively high concentration (Fig. 5). The mean concentrations of Chl $a$, COS, DMS, and CS$_2$ at different depths were 41.7-, 18.9-, 107-, and 37.6-fold higher in summer than spring.

### 3.3 VSCs in the atmosphere

#### 3.3.1 Spring

The concentrations of COS, DMS, and CS$_2$ in the atmosphere overlying the BS and YS in spring were in the range of 255.9–620.2 pptv, 1.3–191.2 pptv, and 5.2–698.8 pptv, respectively (Fig. 6a) and their mean concentrations were





345.6 $\pm$ 79.2 pptv, 47.5 $\pm$ 49.8 pptv, and 113.2 $\pm$ 172.3 pptv. The decreasing order of the three VSCs mean
concentrations in the atmosphere during spring was $COS > CS_2 > DMS$.

**3.3.2 Summer**

The concentrations of COS, DMS, and $CS_2$ in summer ranged from 394.6 to 850.1 pptv, from 10.3 to 464.3 pptv, and
from 15.3 to 672.7 pptv, respectively, with mean values of 570.7 $\pm$ 179.2 pptv, 223.8 $\pm$ 143.6 pptv, and 180.3 $\pm$ 235.6
pptv (Fig. 6b), and the order of the three VSCs in terms of mean concentrations in the atmosphere during summer was
$COS > DMS > CS_2$.

**3.3.3 Seasonal variability of VSCs in the atmosphere**

In spring, the highest concentration of atmospheric COS appeared at station B72 (Fig. 6a), which was located near the
northern Shandong Peninsula. The highest atmospheric DMS concentration appeared at station B08 (Fig. 6a). At
station B49, the DMS concentration in the seawater (Fig. 2) was not as high as that in the atmosphere (Fig. 6a).
According to the 72 h backward trajectory map (Fig. S2), the air mass over station B49 had migrated from the land to
the ocean, passing through Beijing, Tianjin, and other densely populated areas. The lowest atmospheric DMS
concentration appeared at station B47 (Fig. 6a), probably due to the low DMS concentration in seawater (0.5 nmol L$^{-}$
$^1$). In addition, there were high concentrations of $CS_2$ at stations in the BS, such as B57, B60, and B72, and low
concentrations at stations B17 and B21 in the northern YS (Fig. 6a).
The mean concentrations of atmospheric COS, DMS, and $CS_2$ were 1.7-, 4.7-, and 1.6-fold higher in summer than
spring. In summer, the three VSCs in the atmosphere over the BS and YS had similar spatial distributions. At station
B64, both COS and DMS exhibited their highest concentrations (Fig. 6b). The highest concentration of $CS_2$ in summer
appeared at station B49 near the shore, and the lowest concentration appeared far from shore at station H09 (Fig. 6b).
The distributions of $CS_2$ showed an obvious decreasing trend from inshore to offshore (Fig. 6b).

**3.4 Relationships among COS, DMS, and $CS_2$ and environment factors**

A positive correlation was found between COS and DOC in seawater during summer ($P > 0.05$) (Table S4). There
was no significant correlation among the three VSCs in seawater in spring ($P > 0.05$), but there was a significant
correlation between COS and $CS_2$ in the atmosphere ($P < 0.01$) (Table S3). A significant correlation was found



between DMS and $CS_2$ in surface seawater in summer ($P < 0.01$, Table S4). In addition, there was a significant
correlation between atmospheric COS and $CS_2$ in summer ($P < 0.01$, Table S4). The correlations between the three
oceanic VSCs and atmospheric VSCs were not significant ($P > 0.05$, Table S3 and S4).
**3.5 Sea-to-air fluxes of VSCs**
**3.5.1 Spring**
The sea-to-air fluxes of COS, DMS, and $CS_2$ in spring were 0.01–1.59, 0.06–25.40, and 0.002–0.42 µmol m$^{-2}$ d$^{-1}$,
respectively, with averages of 0.46 ± 0.35, 2.99 ± 4.24, and 0.10 ± 0.10 µmol m$^{-2}$ d$^{-1}$ (Fig. 7). The highest COS sea-
to-air flux appeared at station B36, which had a high wind speed (11.3 m s$^{-1}$). In comparison, the lowest COS sea-to-
air flux occurred at station H01, where the minimum wind speed occurred (0.4 m s$^{-1}$); the lowest sea-to-air fluxes of
DMS and $CS_2$ also occurred at station H01 (Fig. 7). The highest DMS and $CS_2$ sea-to-air fluxes appeared at stations
HS4 and B70, respectively, due to high wind speeds and DMS and $CS_2$ concentrations (Fig. 7).
**3.5.2 Summer**
The sea-to-air fluxes of COS, DMS, and $CS_2$ in summer were 0.02–2.47, 0.10–25.44, and 0.003–1.72 µmol m$^{-2}$ d$^{-1}$,
respectively, with the averages of 0.59 ± 0.52, 6.26 ± 6.27, and 0.30 ± 0.32 µmol m$^{-2}$ d$^{-1}$ (Fig. 8). The mean sea-to-air
fluxes of COS, DMS, and $CS_2$ were 1.3-, 2.1-, and 3.0-fold higher in summer than spring. Consistent with their order
in seawater, the order of the sea-to-air fluxes of the VSCs was DMS > COS > $CS_2$. The lowest sea-to-air fluxes of
VSCs in summer occurred at station B05, which had the lowest wind speed (0.4 m s$^{-1}$) and low seawater concentrations
of VSCs. The highest sea-to-air flux of COS and DMS appeared at stations B23 and H14, respectively, coinciding
with high wind speeds and high COS and DMS concentrations in seawater (Fig. 8). The maximum $CS_2$ sea-to-air flux
appeared at station H28, where the concentration of $CS_2$ in seawater was high (Fig. 8).
**4 Discussion**
**4.1 Spatial and depth distributions of three VSCs in seawater**
**4.1.1 Spatial distributions**



The COS concentrations in this study were similar to those in six tidal European estuaries (Scheldt, Gironde, Rhine,
Elbe, Ems, and Loire) (0.22 nmol L$^{-1}$) (Sciare et al., 2002), the DMS concentrations were lower than previous
observations in the BS and YS in autumn (3.92 nmol L$^{-1}$) (Yang et al., 2014), and the CS$_2$ concentrations were lower
than those in the coastal waters off the eastern coast of the United States (0.004–0.51 nmol L$^{-1}$) (Kim and Andreae,
1992). Besides, the VSC concentrations in the seawater of the BS and YS were significantly higher than those in the
oceanic areas such as the North Atlantic Ocean (Simó et al., 1997; Ulshöfer et al., 1995). Zepp and Andreae (1994)
also showed that COS concentrations in nearshore waters were 40-fold higher than those in the open sea waters. The
higher CDOM concentrations in the nearshore waters may be the reason of the difference (Gueguen et al., 2005).
Different production and consumption mechanisms led to the spatial distributions of COS, DMS, and CS$_2$ being
different. DMS and DMSP concentrations are related to the composition and abundance of phytoplankton (Kurian et
al., 2020; Naik et al., 2020; O'Brien et al., 2022). The highest DMS concentrations at station B21 in spring coincided
with high Chl $a$ concentrations (Fig. 2). Low salinities (< 30) appeared at stations of H25, H26, H34, H35, B43, B66,
and B68 due to river water discharge from Yangtze River Estuary, Yellow River and Laizhou Bay were consistent
with high nitrate, Chl $a$, and DMS concentrations (Fig. 3 and S1). High CS$_2$ concentrations in the coastal waters of
the Yellow River estuary and at stations H18, H19, B30 in spring may be due to high CDOM carried by the Yellow
Sea coastal current and Yellow River and terrestrial input. The emergence of high-CS$_2$-value areas in the YS may
have been because the Yangtze River experiences a flood season during summer and large amounts of sediment are
carried into the sea, making the coastal waters of the South Yellow Sea (especially the surface seawater) more turbid,
while the open seas were affected less by the Yangtze River and were more transparent. With higher transparency,
light-induced reactions in water are more likely to occur. The significant correlation between DMS and CS$_2$ in surface
seawater in summer was consistent with the results of Ferek and Andreae (1983) and Yu et al. (2022). DMS in seawater
is mainly derived from the degradation of DMSP, which is released from algal cell lysis (O'Brien et al., 2022).
Moreover, the algae decay increased CS$_2$ emission rate for the reason of degradation of sulfur-containing amino acids
(Wang et al., 2023). The commonality of their sources resulted in a good correlation between DMS and CS$_2$ in seawater.
Xie et al. (1998) pointed out that CS$_2$ has a photochemical production mechanism that is similar to that of COS, and
both are primarily produced by photochemical reactions of thiol-containing compounds such as methyl mercaptan
(MeSH) or glutathione under the catalysis of CDOM. Terrestrial CDOM had higher photochemical reactivity and was
more conducive to the photochemical generation of CS$_2$ (Xie et al., 1998). COS and CS$_2$ are formed via reaction



between cysteine and intermediates (e.g., CDOM˙, ˙OH) (Chu et al., 2016; Du et al., 2017; Modiri Gharehveran et al.,
2020). Modiri Gharehveran and Shah (2021) proposed that DOM can photochemically produce $^3CDOM^*$, $^1O_2$, $H_2O_2$,
and ˙OH via sunlight, which react with DMS and form a sulfur- or - carbon centered radical and then COS and $CS_2$.
High COS concentrations in spring may be due to the influence of the sediment input from the Yellow River into the
BS, which was more turbid and not conducive to the photochemical production of COS. Li et al. (2022) demonstrated
that high nitrate concentration played a key role in high COS production. Two high COS concentrations at stations
H25 and B43 during summer coincided with high nitrate concentration. However, no significant correlations were
found between COS and nitrate concentrations for all the stations during both spring and summer (Tables S3 and S4).
Zepp and Andreae (1994) showed that the photochemical production rates of COS were higher in coastal waters than
in the open sea. Flöck et al. (1997) also believed that the photochemical production of COS benefited from the catalysis
by CDOM. Uher and Andreae (1997) showed that COS in seawater was significantly correlated with CDOM,
indicating that organic compounds affect the concentrations and distributions of COS in seawater. The positive
correlation between COS and DOC in seawater during summer was consistent with Uher and Andreae (1997).

The VSCs in seawater exhibited significant seasonal differences between spring and summer in this study. Xu et al.

(2001) observed that the COS concentrations in the R/V *Polarstern* of South Africa in summer were higher than those
in autumn. In addition, observations by Weiss et al. (1995) showed that the COS concentrations in the seawater of the
Atlantic and Pacific Oceans were very low in winter. In this study, the concentrations of the three VSCs in seawater
during summer were higher than those in spring, which may be due to the higher Chl *a* in summer (mean: 1.60 μg L$^-$
$^1$) than in spring (mean: 1.19 μg L$^{-1}$). Similar to the seasonal changes in Chl *a*, DMS concentrations were higher in
summer than in spring. Indeed, higher phytoplankton biomass in summer has been linked to higher DMS
concentrations in summer than in autumn (Yang et al., 2015). Our results showed that the average concentrations of
COS, DMS, and $CS_2$ in the surface seawater of the BS and YS during summer were higher than those during spring,
and also higher than those in the Changjiang estuary and the adjacent East China Sea (Yu et al., 2022). Different sea
areas, temperature, and industry production may be the difference reasons, and in terms of the latter, the rayon
production which is the main source of anthropogenic $CS_2$ (Campbell et al., 2015) in the northern city of BS.
**4.1.2 Depth distributions**
DMS, COS and $CS_2$ showed similar patterns and decreased with the increasing depth, which agreed with the results
in Yu et al (2022). The highest Chl *a* concentrations during summer occurred at depths of 10–20 m. This was mainly





due to the abundance of nutrients and suitable water temperature near the thermocline, which benefitted phytoplankton
growth. DMS is mainly originated from phytoplankton and coincided with Chl *a* changing pattern. COS and $CS_2$ in
seawater were mainly derived from photochemical reactions of organic sulfides catalyzed by CDOM, so light became
the limiting factor for their production in seawater (Uher and Andreae, 1997). Ulshöfer et al. (1996) studied the depth
distribution of COS in seawater and found that the high concentrations of COS all appeared in the euphotic zone. High
COS concentrations in the surface seawater in spring in this study may be due to the fact that the photochemical
production reactions of $CS_2$ and COS mainly occurred in the euphotic zone because they are dependent on light (Flöck
et al., 1997; Xie et al., 1998). Hobe et al. (2001) argued that non-photochemical production of COS plays an important
role in its global budget. Consistent with Hobe et al. (2001), high COS concentration in the bottom waters of station
H16 in summer may be related to the non-photochemical production of COS or release by underlying sediments. High
$CS_2$ concentrations in the bottom seawater at station H15 in spring may be caused by release from the underlying
sediments. Jørgensen and Okholm-Hansen (1985) found that the release rate of VSCs (such as $CS_2$) in oxygen-
containing surface seawater was usually 10 to 100 times lower than that in underlying sediments in a Danish estuary,
indicating that release from sediments is an important source of $CS_2$. It has been shown that $CS_2$ can be produced by
anaerobic fermentation by bacteria and by reactions between $H_2S$ and organic matter in pore water (and anoxic basins)
(Andreae, 1986). This hypothesis agreed with the results of Wakeham et al. (1987), where the concentration of $CS_2$
peaked (at about 20 nmol $L^{-1}$) near the sediment-water interface.
**4.2 VSCs in the atmosphere**
Similar to our results for VSCs concentrations in the atmosphere during summer, Kettle et al. (2001) found that the
COS concentration in the Atlantic Ocean atmosphere was 552 pptv and Cooper and Saltzman (1993) measured a DMS
concentration of 118 pptv. In addition, the concentrations of atmospheric $CS_2$ in this study were similar to
concentrations of $CS_2$ observed in a polluted atmosphere (Sandalls and Penkett, 1977), but much higher than the $CS_2$
concentration in unpolluted atmospheres such as over the North Atlantic (Cooper and Saltzman, 1993). This indicated
that industrial production and human activities had a significant impact on the concentration of $CS_2$ in the atmosphere.
The mean VSCs concentrations in the atmosphere during summer in this study were all higher than those in the
Changjiang estuary and the adjacent East China Sea (Yu et al., 2022) due to different areas.





The difference in the order of atmospheric VSC concentrations between spring and summer was mainly due to the
high DMS concentrations in the surface seawater during summer. The concentrations of VSCs in the atmosphere
during summer were much higher than in spring, which may be attributed to anthropogenic activities and industrial
emissions. Chin and Davis (1993) showed that anthropogenic sources can significantly affect the concentration of
VSCs in the atmosphere. Anthropogenic VSCs emissions can be evaluated by using isotope measurements (Hattori et
al., 2020). Unfortunately, anthropogenic VSCs emissions were not evaluated in this study, and isotope measurements
will be further studied in the future. In addition, the seasonal changes in the COS, DMS, and $CS_2$ concentrations in
the atmosphere overlying the BS and YS were consistent with those in seawater. Therefore, the high concentrations
of atmospheric COS, DMS, and $CS_2$ during summer were likely associated with high VSCs concentrations in seawater,
which meant that sea-to-air diffusion was an important source of atmospheric VSCs. However, no significance was
found between VSCs in the seawater and atmosphere, which may be because that VSCs in the atmosphere were not
only from sea-to-air diffusion, but also from complex sources such as soil, incomplete burning of biomass, and
industrial releases (Chin and Davis, 1993).
The distributions of atmospheric COS above the BS and YS were relatively consistent with respect to their
concentrations in seawater, which was related to the chemical inertness of COS in the atmosphere (Crutzen, 1976). In
spring, the highest concentration of atmospheric COS at station B72 and DMS at station B08 may coincided with the
anthropogenic emissions and high DMS concentration in seawater, respectively (Fig. 6). The $CS_2$ generated by
industrial activities may influence the atmosphere at station B49 which is near industrial cities like Tianjin. Studies
have shown that anthropogenic input accounts for about 50% of the total $CS_2$ in the global atmosphere, and both
chemical production and pharmaceutical industries are large emitters of $CS_2$ into the atmosphere (Chin and Davis,
1993). $CS_2$ is the main precursor of COS in the atmosphere, and atmospheric $CS_2$ is oxidized to COS by radicals such
as OH with a conversion efficiency of 0.81 (Chin and Davis, 1995). Significant correlation between atmospheric COS
and $CS_2$ in our study demonstrated this.
The 72 h backward trajectory map over station B49 indicated that the high atmospheric DMS concentration was
produced by human activities. The highest concentrations of COS and DMS at station B64 in summer may be caused



by anthropogenic emissions in the coastal region. The highest $CS_2$ concentration in summer at station B49 near
industrial cities like Tianjin may be related to the $CS_2$ generation by industrial activities.
**4.3 Sea-to-air fluxes of VSCs**
The spatial variability in sea-to-air fluxes were consistent with changes in wind speed, which was because sea-to-air
fluxes are dependent on the transmission velocities of VSCs in seawater, which are related to wind speed and viscosity
of seawater, i.e., the highest COS sea-to-air flux at station B36 and the lowest three VSCs sea-to-air flux at station
H01 in spring (Fig. 7). Besides wind speed, the oceanic VSCs concentrations are related to the sea-to-air fluxes, i.e.,
highest DMS and $CS_2$ sea-to-air fluxes at stations HS4 and B70 in spring (Fig. 7). Wind speed was the main influencing
factor compared with the concentration of VSCs in the seawater. Although the oceanic VSCs concentrations at station
H01 were high, the low wind speed reduced the transmission velocity of the VSCs at the sea-to-air interface in this
region, resulting in low sea-to-air fluxes.
In addition, the sea-to-air fluxes of all three VSCs in both spring and summer were positive, indicating that the
seawater was a source of COS, DMS, and $CS_2$ to the atmosphere through sea-to-air diffusion. However, while our
findings agreed with Chin and Davis (1993) and Yu et al. (2022) who showed that the ocean was a major atmospheric
source of COS, they conflicted with Weiss et al. (1995) and Zhu et al. (2019) who found significant COS
undersaturation in some seas. Therefore, the ocean may become a sink of atmospheric COS in some areas or times of
year. The results of this study, which covered offshore and inshore areas, showed that most of coastal and estuarine
seas were sources of COS in the atmosphere, but open seas can have markedly lower concentrations of COS and may
become sinks of COS from the atmosphere under the right conditions. The ocean was the main atmospheric source of
DMS and $CS_2$ due to their low concentrations in atmosphere.
**5 Conclusions**
The distributions of COS, DMS, and $CS_2$ in the surface seawater and marine atmosphere of the BS and YS during
spring and summer exhibited significant spatial and seasonal variability. First, COS, DMS, and $CS_2$ had higher
concentrations in summer than in spring. Second, both COS and $CS_2$ had higher concentrations in coastal waters than
in offshore waters, which may due to higher photochemical reaction rates in nearshore waters compared with the open
sea. In summer, areas with similarly high values of COS and $CS_2$ emerged, indicating that they may have common



photochemical production mechanisms. The depth distributions of COS, DMS, and $CS_2$ were characterized by high
concentrations in the surface seawater that decreased with depth.
In addition, the atmospheric VSC concentrations of the BS and YS exhibited obvious seasonal differences, with
higher concentrations in the summer than in the spring. The VSCs in the atmosphere over the BS and YS had similar
spatial distributions, with declining trends from inshore to offshore areas, especially for $CS_2$. This phenomenon
illustrated the effect of anthropogenic emissions on the atmospheric concentrations of VSCs. Finally, high sea-to-air
fluxes of COS, DMS, and $CS_2$ in the BS and YS indicated that marginal seas were major sources of atmospheric VSCs
and may make considerable contributions to the global sulfur budget.
*Data availability*. Data to support this article are available at https://doi.org/10.6084/m9.figshare.14971644.
*Author contributions.* All authors were involved in the writing of the paper and approved the final submitted paper.
YJ and YL were major contributors to the study's conception, data analysis and drafting of the paper. HZ, LJG and
LQ contributed significantly to writing-original draft. YGP contributed to writing-reviewing, and editing.
*Competing interests*. The authors declare that they have no conflict of interest.
*Acknowledgements*. We are grateful to the captain and crew of the R/V "*Dong Fang Hong 2*" for their help and
cooperation during the in situ investigation.
*Financial support*. This work was funded by the National Natural Science Foundation of China (41976038, 41876122),
and the National Key Research and Development Program (2016YFA0601301).

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



**Figure captions**

**Fig. 1.** Sampling stations in the Yellow Sea and Bohai Sea during (a) spring and (b) summer (▲ indicates stations where atmospheric samples were collected). Yellow Sea Costal Current: YSCC; Changjiang Diluted Water: CDW; Yellow Sea Cold Water Mass: YSCWM; Yellow Sea Warm Current: YSWC; Taiwan Warm Current: TWC; Kuroshio Current: KC. The maps were plotted with Ocean Data View (ODV software) (Schlitzer, 2023).

**Fig. 2.** Spatial distributions of temperature, salinity, Chl $a$, COS, DMS, $CS_2$, and DOC in the surface water of the BS and YS in spring.

**Fig. 3.** Spatial distributions of temperature, salinity, Chl $a$, COS, DMS, $CS_2$, and DOC in the surface water of the BS and YS in summer.

**Fig. 4.** Depth distributions of temperature, salinity, Chl $a$, COS, DMS, and $CS_2$ in seawater in spring.

**Fig. 5.** Depth distributions of temperature, salinity, Chl $a$, COS, DMS, and $CS_2$ in seawater in summer.

**Fig. 6.** Spatial distributions of COS, DMS, and $CS_2$ in the atmosphere over the BS and YS in (a) spring and (b) summer.

**Fig. 7.** Variations of sea-to-air fluxes of VSCs, VSCs concentrations in seawater, and wind speeds in the BS and YS in spring 2018.

**Fig. 8.** Variations of sea-to-air fluxes of VSCs, VSCs concentrations in seawater, and wind speeds in the BS and YS in summer 2018.

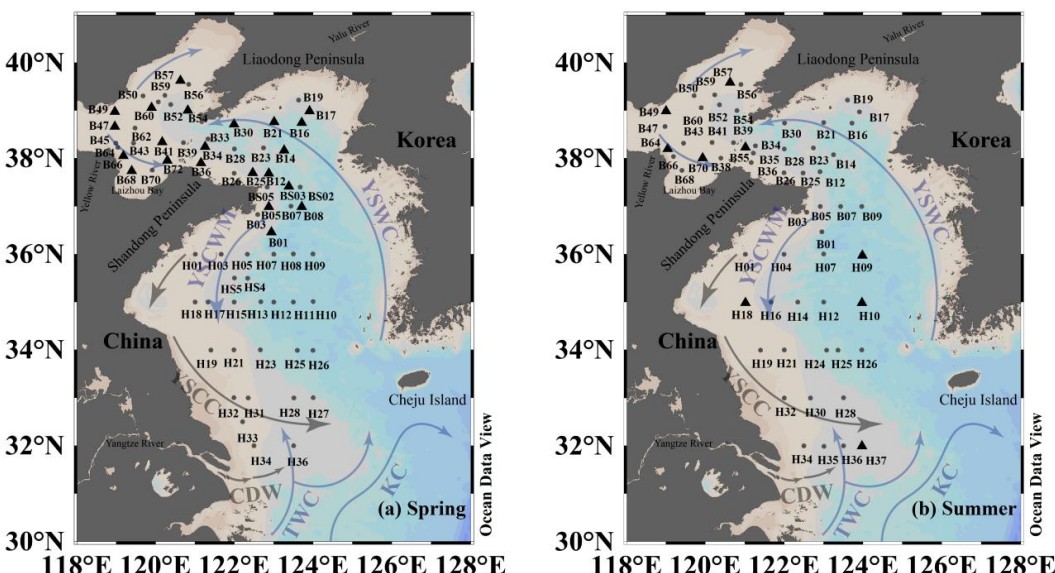

**Fig. 1.** Sampling stations in the Yellow Sea and Bohai Sea during (a) spring and (b) summer (▲ indicates stations

where atmospheric samples were collected). Yellow Sea Costal Current: YSCC; Changjiang Diluted Water: CDW;

Yellow Sea Cold Water Mass: YSCWM; Yellow Sea Warm Current: YSWC; Taiwan Warm Current: TWC;

Kuroshio Current: KC. The maps were plotted with Ocean Data View (ODV software) (Schlitzer, 2023).




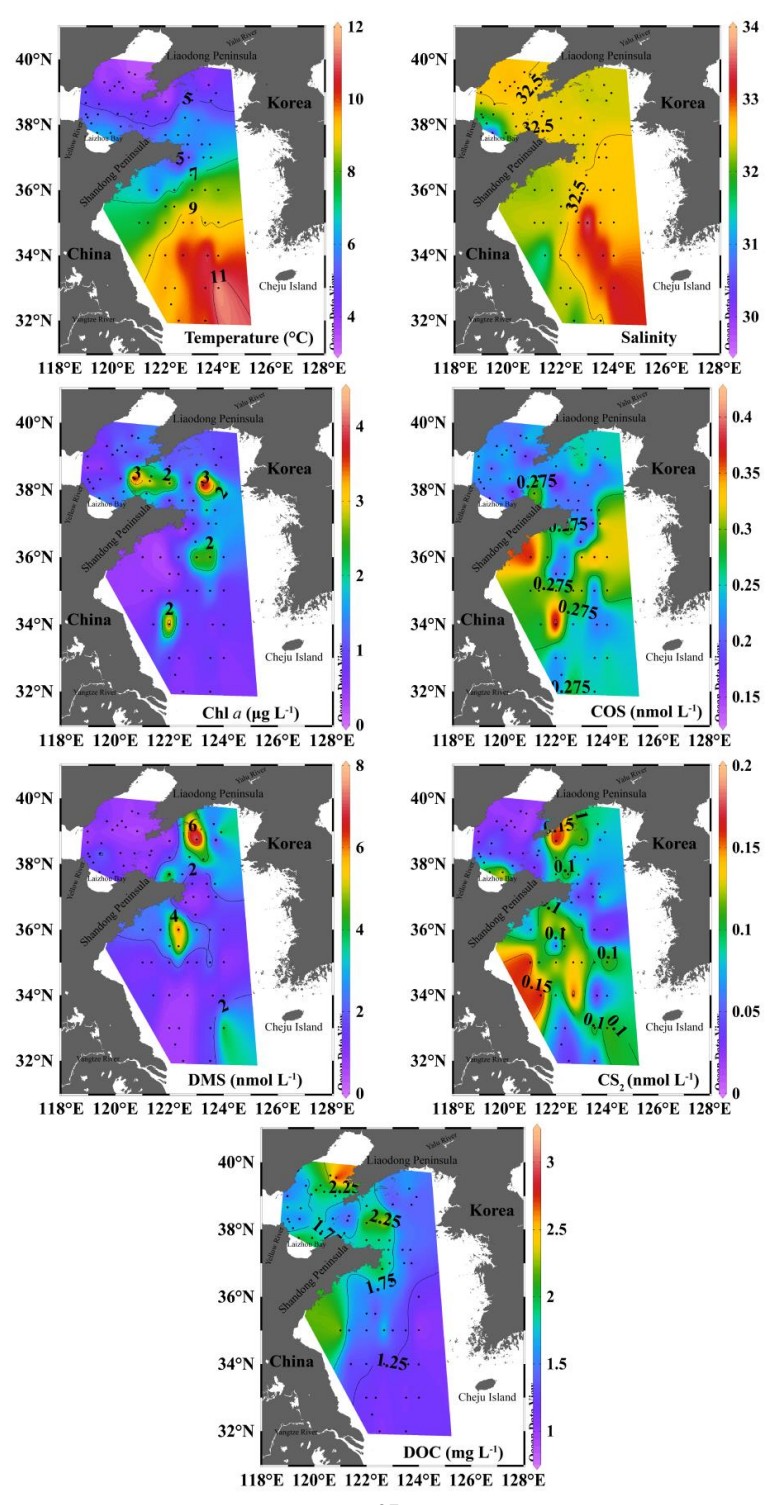




**Fig. 2.** Spatial distributions of temperature, salinity, Chl *a*, COS, DMS, CS$_2$, and DOC in the surface water of the BS
and YS in spring.

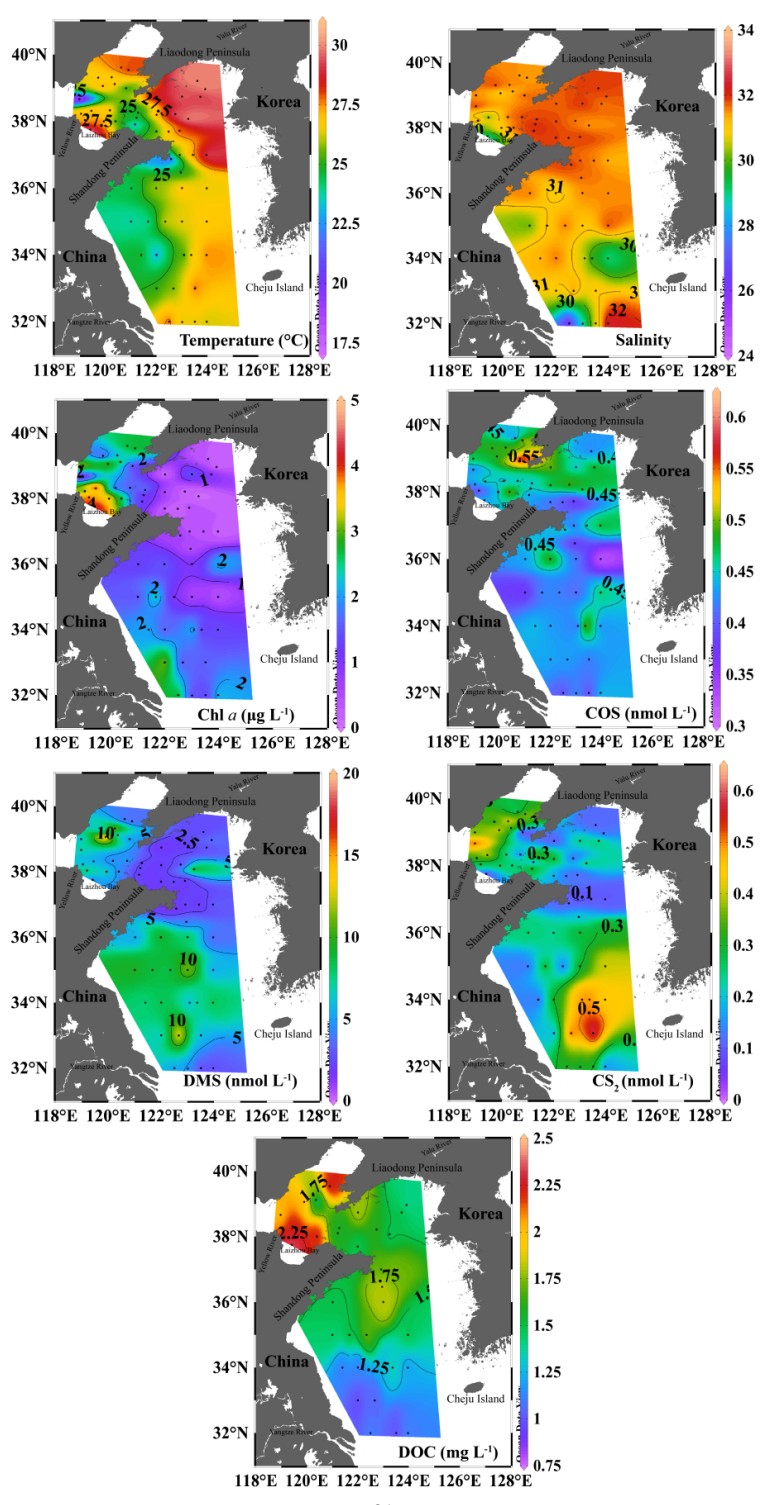




**Fig. 3.** Spatial distributions of temperature, salinity, Chl *a*, COS, DMS, CS$_2$, and DOC in the surface water of the BS

and YS in summer.

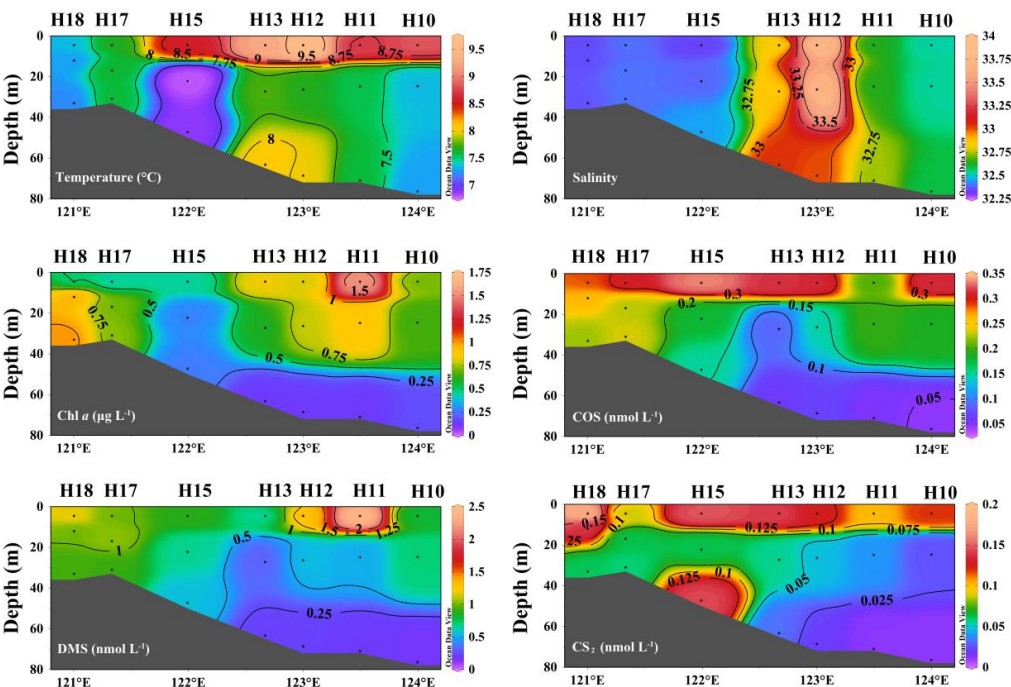

**Fig. 4.** Depth distributions of temperature, salinity, Chl *a*, COS, DMS, and CS₂ in seawater in spring.



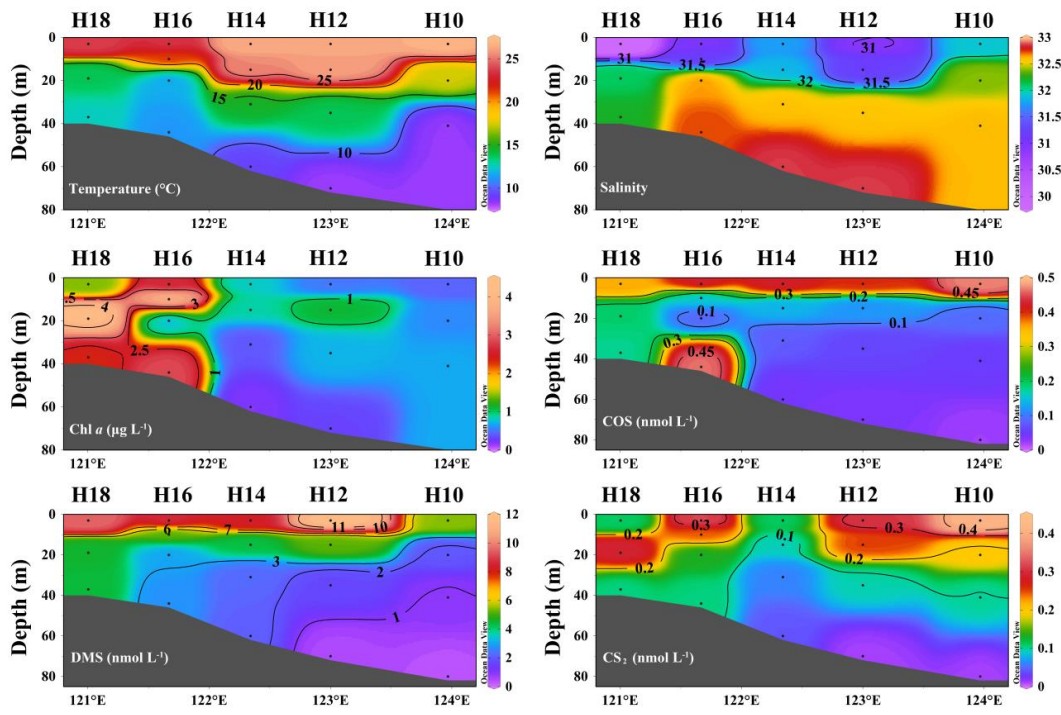

**Fig. 5.** Depth distributions of temperature, salinity, Chl *a*, COS, DMS, and CS$_2$ in seawater in summer.



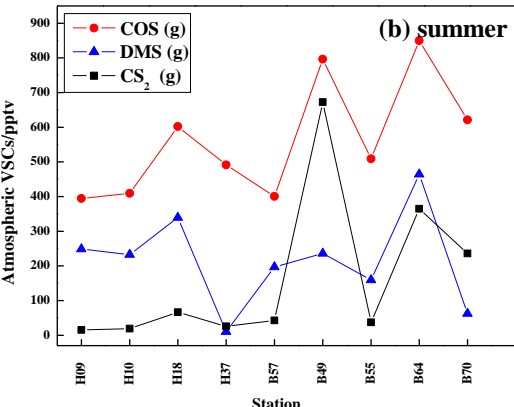



**Fig. 6.** Spatial distributions of COS, DMS, and $CS_2$ in the atmosphere over the BS and YS in (a) spring and (b) summer.



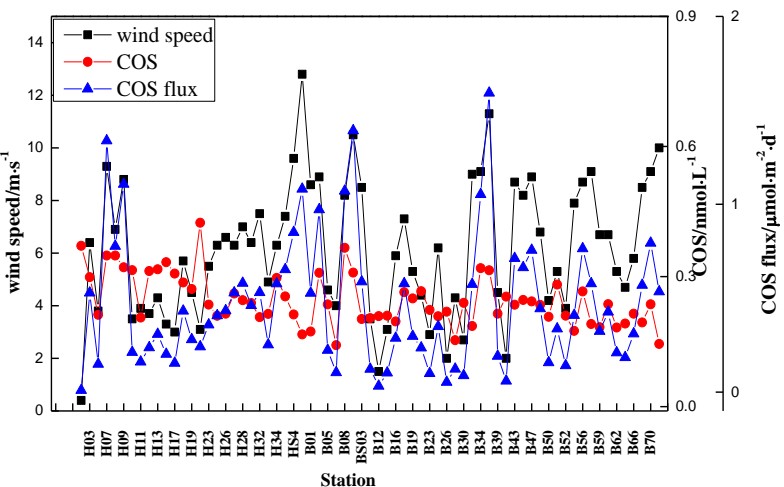


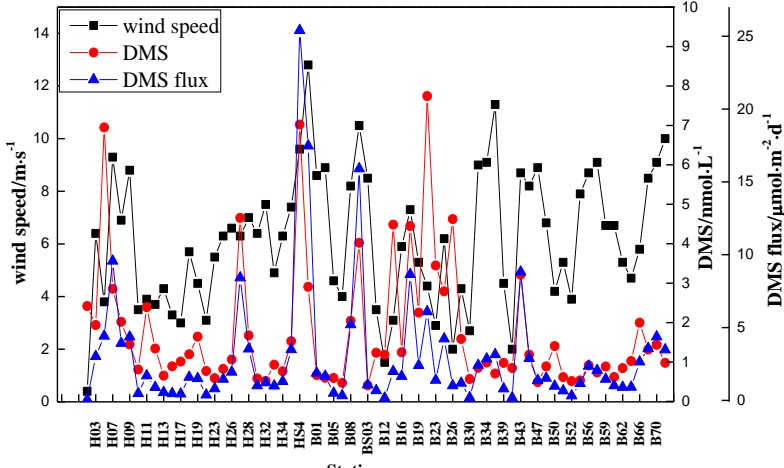






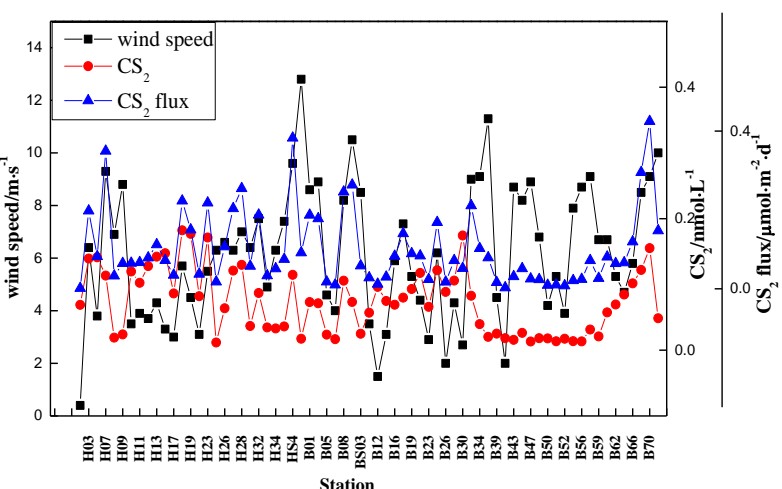


**Fig. 7.** Variations of sea-to-air fluxes of VSCs, VSCs concentrations in seawater, and wind speeds in the BS and YS
in spring 2018.



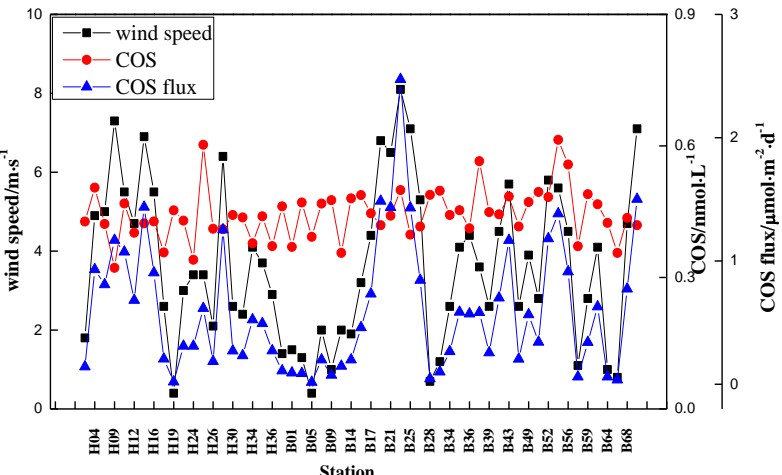


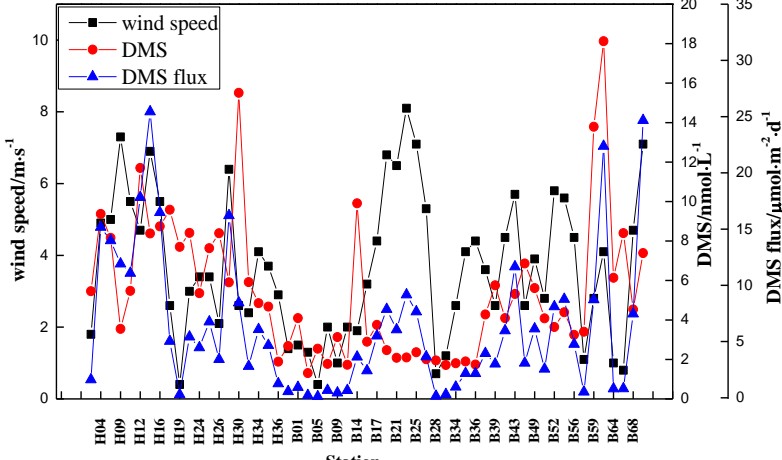


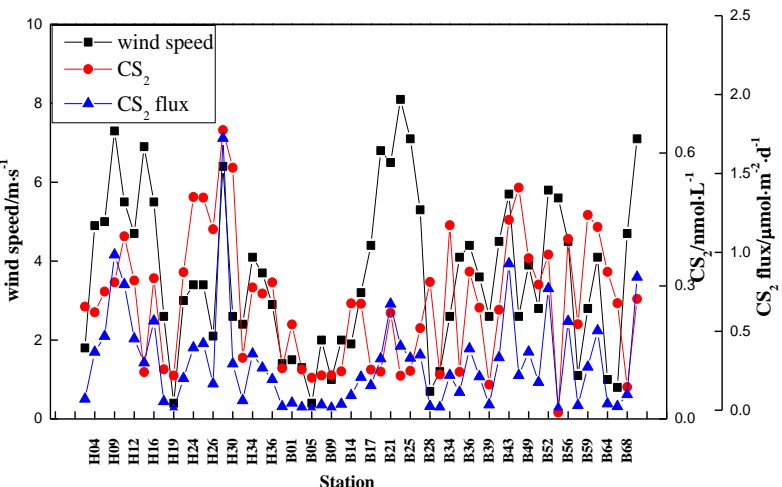


**Fig. 8.** Variations of sea-to-air fluxes of VSCs, VSCs concentrations in seawater, and wind speeds in the BS and YS

in summer 2018.