# Peer review of "Spatial and seasonal variability in volatile organic sulfur compounds in seawater and the overlying atmosphere of the Bohai and Yellow Seas"

_Biogeosciences, 2023_

## Author Response (AR1)

**Response to the referees and the editor**

**Response to the editor**

Thank you for submitting your manuscript to Biogeosciences. Your manuscript has been reviewed by two reviewers. Both reviewers raised major concers about the overall too superficial (too descriptive) presentation of your results and ask for a detailed discussion and better justification of the conclusions.

Thank you for your detailed replies to the reviewers' comments.

Altogether, I can recommend re-submission of the manuscript only after major revisions. When revising the manuscript, please, pay special attention to address the major concern of the two reviewers: 'However, the scientific content of the manuscript remains pretty descriptive.' (rev#1) and 'What are the major findings from this work?' (rev#2).

I am looking forward to the revised manuscript.

Yours Sincerely
Hermann Bange

Response: Thank you for the comments of the editor. According to the two referees' comments, some descriptive discussions have been deleted, we have focused the findings, and give a detailed discussion and better justification of the conclusions in the revised manuscript.

**Response to the referee #1**

In the manuscript "Spatial and seasonal variability in volatile organic sulfur compounds in seawater and overlying atmosphere of the Bohai and Yellow Seas" Yu et al., compare surface measurements and depth profiles of marine OCS, DMS and $CS_2$ in two different seasons (spring and summer). Accompanied by ancillary data (ocean temperature, salinity, chla, nitrate, DOC) the authors try to interpret their data related to production and loss processes of each sulfur compound. Finally, using also atmospheric OCS, DMS and CS2 measurements they calculate the sea-to-air-flux of the described sulfur compounds.

Measurements of sulfur compounds in the ocean and atmosphere are scarce (especially $CS_2$ and OCS in comparison to DMS), but they are urgently needed to investigate their influence on a global scale. Therefore, this dataset is a valuable contribution to increase the number of measurements during different seasons in this specific marginal sea area. However, the scientific content of the manuscript remains pretty descriptive. The discussion part seems very comprehensive but at the same time stays superficial. The introduction part ends with "…we investigate…variability of COS, DMS, and $CS_2$…to better understand production and loss processes of VSCs". Here, I strongly disagree. The authors know and also mention in the introduction the different parameters (e.g. CDOM, DMSP, bacteria) which influence (photochemical or biological) production and loss of the presented sulfur compounds but this ancillary data is not presented here.

I suggest to revise the manuscript following the main comments below, also with respect to the English language, before publication.

Response: The influences on a global flux have been evaluated.

The methods have been added some description, and some deep discussion has been added. The sentence in the introduction part has been changed.

The English language has been edited by a professional language editing service-EditorBar Language Editing. The certificate of language editing is shown in the last page.

Some superficial and descriptive discussion has been deleted, i.e., "Thus, the highest COS and lowest sea-to-air fluxes occurred at station B36 and at station B12, respectively, in spring (Fig. 7). Besides the wind speed, the sea-to-air fluxes are related to oceanic VSC concentrations, i.e., the highest DMS and $CS_2$ sea-to-air fluxes were observed at stations HS4 and B68 in spring (Fig. 7).", and "Although the oceanic VSC concentrations were high at station H01, the low wind speed reduced the transmission velocity of the VSCs at the sea-to-air interface in this region, resulting in low sea-to-air fluxes.", and "The results of this study, which covered offshore and inshore areas, showed that most coastal and estuarine seas were sources of COS in the atmosphere. However, open seas can have markedly lower concentrations of COS and may become sinks of COS from the atmosphere under the right conditions. The ocean was the main atmospheric source of DMS and $CS_2$ due to their low concentrations in the atmosphere." in Section 4.3, and "The high $CS_2$ concentrations in the YS may be attributed to the Yangtze River flood season during summer, when large amounts of sediment are carried into the sea, increasing the turbidity of the coastal waters of the South YS (especially the surface seawater). In contrast, the open sea areas were less affected by the Yangtze River and had lower turbidity; thus, light-induced reactions in water were more likely." in section 4.1.1, "the COS and $CS_2$ concentrations were higher in coastal waters than in offshore waters, which may be due to higher photochemical reaction rates in nearshore waters than in the open sea. Areas with similar high COS and $CS_2$ concentrations were observed in summer, indicating that a similar photochemical production mechanism occurred.", "The VSCs in the atmosphere over the BS and YS had similar spatial distributions, with declining trends from inshore to offshore areas, especially for $CS_2$. This result highlighted the effect of anthropogenic emissions on the atmospheric mixing ratios of VSCs." in conclusion, et al.

General comments

Introduction

The introduction should be clearly structured. Presentation of different production and loss processes is mixed for COS, DMS and $CS_2$. It would help the flow to clearly distinguish between these three compounds and their production/loss processes.

Response: The production and loss processes COS, DMS, and $CS_2$ have been shown in different paragraph and clearly distinguish between these three compounds.

Material and Methods

The sampling/measurement procedure of the ancillary data (section 2.4) should be presented in a bit more detail. Also, phosphate and silicate measurements are missing in this section, although data is presented in Table S3 and Table S4.

Response: The sampling/measurement procedure of the ancillary data (section 2.4) has been presented in a bit more detail. Also, phosphate and silicate measurements have been added in this section.

Discussion

The authors explain parts of their results and also relate their results to other findings. However, some parts should go in to the introduction part as this is state-of-the-art knowledge. This would also give the introduction a more detailed content, also with respect to the findings of this study.

Response: Some parts have been added in the introduction part. See also, "COS production is dependent on UV radiation, chromophoric dissolved organic matter (CDOM), cysteine, and nitrate concentration (Lennartz et al., 2021; Li et al., 2022). COS production rates increase with increasing nitrate concentration (Li et al., 2022).", "$^3CDOM^*$, $^1O_2$, $H_2O_2$, and $^{\bullet}OH$ produced by the photochemical reaction of DOM react with DMS and produce COS and $CS_2$ (Modiri Gharehveran and Shah, 2021).".

Oceanic COS is known to have a distinct seasonal, but also diurnal cycle due to the photochemical production. This is not at all mentioned or discussed in the manuscript, especially with respect to the different times of samplings (spring to summer but also potentially on a diurnal basis.

Response: Seasonal and diurnal variations of COS discussion has been added in section 4.1.3 "4.1.3 Seasonal and diurnal variations in VSCs in seawater".

I was missing the main story in the discussion part. The authors relate their findings to some other studies in the same area also with respect to different seasons which is good and valuable. However, what is about the bigger picture or how can the results from the YS and BS be referred to other marginal seas? The authors highlight the influence of oceanic sulfur emissions on the atmospheric chemistry. How strong are emissions of those compounds compared to other regions and on global scale? The authors state in the conclusion "marginal seas…make a considerable contribution to the global sulfur budget" but miss to discuss and prove this with actual numbers. The DMS climatology from Hulswar et al. (2022) (not even cited) or a compilation of $CS_2$ and COS measurements by Lennartz et al. (2020) could help as a start to discuss the findings in a global context.

Response: Hulswar et al. (2022) has been cited. The following sentences about global fluxes have been added in the discussion section 4.3.

The model of Lennartz et al. (2021) was not used to evaluate the global sea-air fluxes of DMS, OCS, $CS_2$ in this study due to a lack of parameters, i.e., the absorption coefficient of CDOM at 350 nm (a350), global radiation (converted to UV radiation), and sea surface pressure. Therefore, the global sea-air fluxes of DMS were calculated following Hulswar et al. (2022) with minor modifications. The global sea-air fluxes of

OCS or $CS_2$ were evaluated by the mean sea-air fluxes of OCS or $CS_2$ multiplied by the ocean area and the time. The global sea-air fluxes of DMS, OCS, and $CS_2$ were 21.3, 2.3, and 2.0 TgS $year^{-1}$, respectively. The global sea-air flux of DMS was similar to the results of Hulswar et al. (2022) (27.1 TgS $year^{-1}$). In comparison, the global sea-air fluxes of OCS and $CS_2$ were 15.9- and 9.9-fold higher than the results of Lennartz et al. (2021). The different calculation method we used may overestimate the global sea-air fluxes of OCS and $CS_2$. The another reason may be the high sea-air fluxes of OCS or $CS_2$ in the BS and YS because marginal seas are significantly influenced by anthropogenic emissions (Watts, 2000). The sea-air fluxes of DMS, OCS, and $CS_2$ in the BS and YS were 28.2, 3.1, and 2.7 GgS $year^{-1}$, accounting for 0.10%, 2.23%, and 1.44% of global sea-air fluxes. The BS and YS comprise 0.13% of the global sea area; therefore, they contribute considerably to global sea-air fluxes.

Specific comments

ll.39: "Some researches indicates that the ocean is the source of VSCs. Opposite results also were reported that the ocean is the sink of VSCs." I do not think that this is true for DMS and CS2. In case the authors relate this to COS (as the citation suggests), please revise this sentence to make it COS specific.
Response: The sentence has been changed into "Some studies have indicated that the ocean is a COS source (Chin and Davis, 1993; Yu et al., 2022), whereas others have shown that the ocean is a COS sink (Zhu et al., 2019).".

ll.57: "The production and loss of VSCs involves in phytoplankton and bacteria synthesis, zooplankton grazing, bacterial degradation, sea-air diffusion, photo-oxidation and/or photochemical reaction". This is a very general sentence. Please be more precise with respect to the different compounds presented in the manuscript.
Response: The sentence has been changed into "The production and loss of DMS involve phytoplankton and bacteria synthesis, zooplankton grazing, bacterial degradation, and sea-air diffusion (Schäfer et al., 2010). COS and $CS_2$ production are related to photo-oxidation and/or photochemical reactions (Lennartz et al., 2020; Xie et al., 1998).".

ll.68: "In this study, we investigate… the effects of YSCWM on VSCs distributions to better understand the production and loss processes of VSCs." As already mentioned I think this sentence is too ambitious with respect to the dataset.
Response: The sentence has been changed into "…and the effects of the YSCWM (the 35 °N transect) on the VSC distributions to better understand the distributions and impact factors of VSCs in Chinese marginal seas.".

l.98: "Based on the similarities…" I guess the authors want to say that they calculated the concentrations with help of a calibration using standard gases?

Response: Yes, the reviewer is right. The sentence has been changed into "The VSC concentrations were calculated after calibration using standard gases (Fig. S1).".

l.110: "The detection limit of the method for VSCs was 2.5-3.5 ng…" According to section 2.2 the authors used 30mL of sample to measure COS, DMS and $CS_2$ in seawater. Using this volume and a detection limit of 2.5ng would result in a detection limit concentration of ~1.3nmol/L. However, most of the presented DMS data and all of the presented $CS_2$ and COS data falls below this threshold. Please check.
Response: The original detection limit is wrong. We have checked the data and the sentence has been changed into "The detection limits of the method for COS, DMS, and $CS_2$ were 33 pg, 387 pg, and 22 pg and the measurement precision was 5.59%-11.70% (Tian et al., 2005).".

ll.120: "...and selected ion monitoring mode (SIM)." What masses did the authors use for qualification and quantification of the different compounds?
Response: "The mass-to-charge ratios ($m/z$) for COS, DMS, and $CS_2$ were 60, 62, and 76, respectively", which has been added in 2.2.

ll.161: "The distribution of $CS_2$…(Fig. 2)…was similar with that of DOC." I do not see that.
Response: ", which was similar with that of DOC" has been deleted.

l.169: "...which may have been due to the abundance of nutrients…" Please also show nitrate in both summer and spring figures and not only in the supplement.
Response: The nutrient data were provided by the open research cruise, see 2.4, therefore, it is unsuitable to show the figures in the main text. We show the figures in the supplement to avoid repeating presentation in the main text from others. Figures of phosphate and silicate have been added in Figure S2, and the sentence has been changed into "which may have been due to the abundance of nutrients (nitrate: 5.85 µmol $L^{-1}$, silicate: 17 µmol $L^{-1}$)".

ll.201: "However, in the bottom waters of station H16, COS had a relatively high concentration (Fig. 5)." What means relatively? Please be precise with respect to the actual concentration or with respect to the sampling location the authors compare to.
Response: Thank you for your advice, the word "relatively" used here was Chinese English expression, and it has been deleted and the actual concentration was shown as "the COS concentration was high in the bottom waters of station H16 (0.465 nmol $L^{-1}$)".

ll.202: "The mean concentrations of Chl a, COS, DMS, and $CS_2$ at different depths were … higher in summer than spring." It is not clear by "different depths" what numbers are related to each other.
Response: The mean concentrations of Chl a, COS, DMS, and $CS_2$ of the whole values at different depths were calculated and shown in the data, the original calculated data

were wrong, they have been revised as "The mean concentrations of Chl $a$, COS, DMS, and $CS_2$ of all samples at different depths were 1.2-, 0.0-, 4.6-, and 1.0-fold higher or equal to those in summer (1.34 μg $L^{-1}$, 0.20 nmol $L^{-1}$, 4.38 nmol $L^{-1}$, and 0.158 nmol $L^{-1}$, respectively) than in spring (0.61 μg $L^{-1}$, 0.20 nmol $L^{-1}$, 0.78 nmol $L^{-1}$, and 0.080 nmol $L^{-1}$, respectively).".

Section 3.3.3: The title is misleading and results shown in this section should be moved to section 3.3.1 and section 3.3.2 to add more content to the respective sections.
Response: The title of 3.3.3 has been deleted and the results related to spring and summer shown in this section have been moved to section 3.3.1 and section 3.3.2 respectively to add more content to the respective sections.

l.219 and Fig S2: "According to 72h backward trajectory…". Is there a reason why the authors started the trajectories at 500m, 1000m, and 1500m height? Do the authors have information about the marine boundary layer height? Otherwise I would suggest to start these trajectories at a much lower height in relation to the height of the actual measurements.
Response: Thanks for the suggestion of the reviewer. The original 72h backward trajectory is indeed at too high heights. The trajectories have been redrawn and with a much lower height (10 m, 50 m, and 200 m) in relation to the height of the actual measurements. See Fig. S3.

[Figure]

B08-spring                                  B47-spring

[Figure]

B49-spring                    B49-summer

B64-summer                    H09-summer

Figure S3. 72 h backward trajectory of the air mass above stations B08, B47, B49 in spring and stations B49, B64, H09 in summer in the BS and YS of 2018.

ll.220: "The lowest atmospheric DMS concentration appeared at station B47 (Fig. 6a), probably due to the low DMS concentration in seawater (0.5 nmol L$^{-1}$)." I was

wondering, why the authors only check the backward trajectories once for a single station and not for the whole area? Especially as B49 (backward trajectory provided, high atm DMS) and B47 (no backward trajectory provided, low atm DMS) are very close to each other.

Response: Backward trajectory of stations B49, B47, B08 in spring and B49, B64, H09 in summer have been redrawn to find the sources and the reasons of different VSCs mixing ratios. See Fig. S3.

l.230: "P>0.05" should be "P<0.05".

Response: Yes, the reviewer is right. "P>0.05" has been changed into "P<0.05" in the section 3.4.

section 3.4: Please structure this section logically.

Response: Section 3.4 has been structured logically as follows "A significant correlation was found between the DMS and $CS_2$ concentrations in the surface seawater in spring ($P < 0.05$) and summer ($P < 0.01$) (Table 1). A positive correlation occurred between the COS and DOC concentrations in seawater ($P < 0.05$) and between the $CS_2$ and Chl $a$ concentrations in seawater ($P < 0.05$) during summer (Table 1). There was a significant correlation between the atmospheric COS and $CS_2$ mixing ratios in spring and summer ($P < 0.01$, Table 1).".

ll.300: "In this study, the concentrations of the three VSCs in seawater during summer were higher than those in spring, which may be due to the higher Chl a in summer than in spring." As already outlined in the manuscript, the three VSCs have different sources. Therefore, high chla as a general reason, seems a bit misleading.

Response: According to the comments, the sentence "In this study, the concentrations of the three VSCs in seawater during summer were higher than those in spring, which may be due to the higher Chl $a$ in summer (mean: 1.60 µg $L^{-1}$) than in spring (mean: 1.19 µg $L^{-1}$)." has been changed into "The significant positive correlations between the $CS_2$ and Chl $a$ concentrations during summer may explain the higher $CS_2$ concentration in seawater during summer than during spring in this study." in section 4.1.3.

ll.370: "Wind speed was the main influencing factor…" Did the authors do any statistical analysis?

Response: According to the formula $F = k_w(c_w - c_g/H)$, where $F$ is the sea-to-air flux of VSCs (µmol $m^{-2}$ $d^{-1}$); $k_w$ is the VSCs transfer velocity (m $d^{-1}$); $k_w$ was calculated from wind speed and sea-surface temperature by the N2000 method (Nightingale et al., 2000), Therefore, "wind speed was the main influencing factor…". Statistical analysis has been done, and added "A significant correlation was found between the sea-to-air fluxes of COS, DMS, and $CS_2$ and the wind speeds ($P < 0.01$)." in section 3.5.1, "A significant correlation was found between the sea-to-air fluxes of COS, DMS, and $CS_2$ and the wind speeds ($P < 0.05$)." in section 3.5.2.

Figure 1: Only YSCWM is mentioned in the manuscript. To increase readability of the figure please delete all other current names.

Response: The other currents names have been deleted from Figure 1.

[Figure]

Fig. 1. Sampling stations in the Yellow Sea and Bohai Sea during (a) spring and (b) summer (▲ indicates stations where atmospheric samples were collected). Yellow Sea Cold Water Mass: YSCWM. The maps were plotted with Ocean Data View (ODV software) (Schlitzer, 2023).

Figure 6: Stations are presented in alphabetical order. However, in the manuscript, atmospheric measurements are often related to inshore or offshore locations. It would be great if this information could also be part of this figure for a better comparison and interpretation of the data. Both subplots next to each other and on the same y scale would improve comparability between spring and summer.

Response: To improve comparability, the atmospheric data have been drawn in ODV figures with black circles showing the values, and the inshore or offshore locations can be seen clearly. See Fig. 6.

[Figure]

**Fig. 6.** Spatial distributions of COS, DMS, and $CS_2$ in the atmosphere over the BS and YS in (a)-(c) spring and (d)-(f) summer. (Unit: pptv)

Figure 7 and 8: There are much more datapoints for the fluxes than atmospheric measurements? How is this possible? Are there atmospheric measurements missing in Figure 6?

Response: The original fluxes of COS and $CS_2$ were calculated using the mean atmospheric concentration, and DMS fluxed were calculated with DMS in ocean because the DMS concentrations in the atmosphere are much lower than those in the seawater. The DMS concentrations in the atmosphere can be considered as 0. Therefore, the DMS fluxes are not changed. The fluxes of COS and $CS_2$ have been revised and calculated using the formula $F = k_w(c_w - c_g/H)$ in section 2.3, and the Figure 7 and 8 have been redrawn as follows.

[Figure]

[Figure]

[Figure]

**Fig. 7.** Variations in sea-to-air fluxes of VSCs, VSCs concentrations in seawater, and wind speeds in the BS and YS in spring 2018.

[Figure]

Fig. 8. Variations in sea-to-air fluxes of VSCs, VSCs concentrations in seawater, and wind speeds in the BS and YS

TableS2: Please add references to temperature dependent Henry constants.

Response: The references to temperature dependent Henry constants (De Bruyn et al.,

1995; Dacey et al., 1984) have been added.

Reference:

Dacey, J. W. H., Wakeham, S. G., and Howes, B. L.: Henry's law constants for dimethylsulfide in freshwater and seawater, Geophys. Res. Lett., 11, 991–994, https://doi.org/10.1029/GL011i010p00991, 1984.

De Bruyn, W. J., Swartz, E., Hu, J. H., Shorter, J. A., Davidovits, P., Worsnop, D. R., Zahniser, M. S., and Kolb, C. E.: Henry's law solubilities and Šetchenow coefficients for biogenic reduced sulfur species obtained from gas-liquid uptake measurements, J. Geophys. Res.-Atmos., 100, 7245–7251, https://doi.org/10.1029/95JD00217, 1995.

**Response to the referee #2**

This manuscript is the second from these exact set of cruises to the Yellow and Bohai Seas, by the same authors. Here we are shown the methods for observing COS, CS2, and DMS (DMS is also in the other publication submitted to JGR) and their distributions (horizontal and vertical). Air and water values of the gases were measured and air-sea fluxes computed. Certain factors deemed relevant are correlated with the measured values to understand sources and sinks of these gases in the air and water. This manuscript requires a major overhaul before it can be published. The English needs to be thoroughly revised and the main ideas need to be clearer. What are the major findings from this work? Although the measurements are valuable, in order for them to be published in a scientific journal, there needs to be some insight or something new found. How does this contribution further our understanding? In addition, I am not sure if it is appropriate to publish the DMS values here without citing the other article that has been written about them (I was a reviewer of that article as well). Related to that point, other sections of the article should not be direct copies of the other manuscript submitted about this cruise (methods, etc.). Please check that.

Response: Our manuscript had been edited by a professional language editing service-EditorBar Language Editing. See the revised manuscript. The certificate of language editing is shown in the last page.

The major findings of this work are the seasonal variations in VSCs, distributions of VSCs and the impact factors, the sources of atmospheric VSCs based on the 72 h back trajectories, and the contribution to the global scale.

Yes, this manuscript had been rejected by the journal of JGR-Oceans before, and we have revised the manuscript according the comments of the reviewers. The spatial and depth distribution values of DMS have been cited from the Zhang et al. (2023, JGR-Oceans). Other sections (methods, etc.) of the article have been checked and is not directly copied from the other manuscript submitted about this cruise. The figures about DMS are drawn by ourselves.

Specific comments:

General – Did the authors measure dissolved $O_2$ concentrations? This would be useful information to show, especially for the depth profiles. Also, when discussing the atmospheric values, it would be more proper to call them mixing ratios and not concentrations.

Response: No, we have not measured the dissolved $O_2$ concentrations. Dissolved $O_2$ concentrations is useful information, unfortunately, it is not design in that cruise. We will measure it in the future research.

Thank you for your advice. The atmospheric values have been changed to call them mixing ratios.

Lines 54-55 - Citation formatting is awkward.

Response: The citation formatting has been revised as "Two different approaches (ice core and isotope measurements) were used to evaluate anthropogenic COS emissions (Aydin et al., 2020; Hattori et al., 2020).".

Lines 83-85 – These two sentences can be merged into one.

Response: These two sentences have been merged into one "The stability of VSCs in fused silica-lined canisters has been verified during storage for 16 d at room temperature (Brown et al., 2015).".

Section 2.2 – Why were different instruments used for the air and water measurements? The description of the atmospheric calibration is not clear, specifically regarding the primary standard. It seems like the primary standard was bought and it contained a 1 ppt mixing ratio for all three gases. Is this 1 part per trillion or part per thousand. I understand ppt = part per trillion. If so, this is a very low standard. It would also be nice to see some of the data from the calibrations, and perhaps some schematics of how the instruments were set up, in the supplemental material.

Response: A gas chromatograph (GC) can be used to measure oceanic VSCs. In comparison, the VSCs concentrations in the atmosphere is too low that they can not be measured by a GC, therefore, we used a gas chromatograph-mass spectrometer (*GC-MS*) to measure atmospheric VSCs.

Standard gases were bought and it contained a 1ppmv mixing ratio for all three gases. The sentences in section 2.2 have changed into "Standard VSC gases with mixing ratios of 1 ppmv were bought from Beijing Minnick Analytical Instrument Equipment Center. Qualitative analysis was conducted by comparing the results with the retention times of the standards, and quantitative analysis was conducted by diluting the VSC standard gases to 1 ppbv and 5 ppbv using a 2202A dynamic dilution meter (Nutech, USA) and injecting different volumes of the diluted VSC standards into the GC using a gas-tight syringe. The VSC mixing ratios were calculated after calibration using standard gases (Fig. S1).".

The VSCs standard curves were made as followed:

1. VSCs standard curves in spring

(1) COS standard curve:

The 5 ppbv standard gas was used, and the injection volumes were set as 5, 10, 20, 50, 100 mL. We use the standard gas mixing ratio * injection volume (25, 50, 100, 250, 500) as the X-axis, and the peak area detected as Y-axis. The mixing ratios of COS were calculated according to the peak area and correlative equation ($y = 4008.5x + 371580$). The injection volume of atmospheric gas is 200 mL.

[Figure]

(2) DMS standard curve:

The 1 ppbv standard gas was used, and the injection volumes were set as 0.2, 10, 20, 30, 50, 70 mL. We use the standard gas mixing ratio * injection volume (0.2, 10, 20, 30, 50, 70) as the X-axis, and the peak area detected as Y-axis. The mixing ratios of DMS were calculated according to the peak area and correlative equation ($y = 1976.6\ x - 11.126$).

[Figure]

(3) $CS_2$ standard curve:

The 1 ppbv standard gas was used, and the injection volumes were set as 1, 2, 10, 20, 30, 50, 100 mL. We use the standard gas mixing ratio * injection volume (0.2, 10, 20, 30, 50, 70) as the X-axis, and the peak area detected as Y-axis. The mixing ratios of $CS_2$ were calculated according to the peak area and correlative equation ($y = 17125x + 98420$).

[Figure]

2. VSCs standard curves in summer

(1) COS standard curve:

[Figure]

(2) DMS standard curve:

[Figure]

(3) $CS_2$ standard curve:

[Figure]

3. The schematics of the instruments were set up as follows:

[Figure]

Fig. S1 The VSCs standard curves and the apparatus diagram used for analysis of

VSCs in atmosphere

Section 3.3.3 – There is discussion of atmospheric sources here and some use of back trajectories (supplemental material), but I do not understand why only one station was examined in this way. I think back trajectories from various parts of the cruise track would be extremely useful. The atmospheric lifetimes of the gases are very different, so the back trajectories over multiple timescales for the various regions could tell a different story for each gas.
Response: Backward trajectory of stations B49, B47, B08 in spring and B49, B64, H09 in summer have been redrawn to find the sources and the reasons of different VSCs concentrations. See Figure S3.
72-hour back trajectories mean trajectories from 72 h to 0 h before sampling, therefore, which include 24 h and 48 h (1/3 and 2/3 of the line near the sampling station).

[Figure]

B08-spring                                        B47-spring

[Figure]

B49-spring                                        B49-summer

[Figure]

B64-summer                   H09-summer

Figure S3. 72 h backward trajectory of the air mass above stations B08, B47, B49 in spring and stations B49, B64, H09 in summer in the BS and YS of 2018.

Section 3.4 and supplemental tables – There is no good explanation in the subsequent discussion (section 4) about why the correlations between the different factors change so much, especially between variables such as COS and DOC in seawater.

Response: The original first sentence in section 3.4 is wrong according to table S4, and it has been changed into "A positive correlation occurred between the COS and DOC concentrations in seawater ($P < 0.05$) and between the $CS_2$ and Chl a concentrations in seawater ($P < 0.05$) during summer (Table 1).". The discussion about COS and DOC in seawater has been added in the second paragraph of section 4.1.1. The other correlations have also been discussed in the discussion.

Section 4.1.1 – This seems like a random assortment of statements. What are the main ideas of each paragraph? I had a hard time finding the clear points here.

Response: The Section 4.1.1 mainly stated spatial distributions of VSCs and compared with the other sea areas and the impact factors. The first paragraph states the spatial distributions of VSCs in this study and analysis the results in others' studies. The second paragraph states the reasons, the impact factors, and production or consumption which resulted in the spatial distributions. We have modified the structure and put the photochemical mechanisms of CDOM together and delete the wordy sentences: "High COS concentrations in spring may be due to the influence of the sediment input from the Yellow River into the BS, which was more turbid and not conducive to the photochemical production of COS.", and "The high $CS_2$ concentrations in the YS may

be attributed to the Yangtze River flood season during summer, when large amounts of sediment are carried into the sea, increasing the turbidity of the coastal waters of the South YS (especially the surface seawater). In contrast, the open sea areas were less affected by the Yangtze River and had lower turbidity; thus, light-induced reactions in water were more likely.". The third paragraph stating seasonal and diurnal variations of VSCs has been moved to section "4.1.3 Seasonal and diurnal variations of VSCs in seawater". Discussions about seasonal and diurnal variations of VSCs have been added according to the advices of the other referee #1.

Section 4.1.2 – I again do not understand the point of this section. What is new? The information cited is very old. Yes, COS and $CS_2$ processes depend on light. What is added here? Also, the statements at the end of the paragraph about sulfur in the deeper sea cannot be substantiated, as no dissolved oxygen measurements are presented. Finally, the Lennartz et al. ESSD database paper is cited, but was it used in any way to put the measurements in some context? The data presented in this manuscript should also be submitted to that database. This would be a wonderful way to use this data (for COS, $CS_2$, air and water). There was a follow-on paper in ESSD (Lennartz et al., 2021) that looked more deeply into modelling gas exchange and a separate Lennartz et al. (2019) publication on oceanic processes. These might be useful to consider as well.

Response: The point of section 4.1.2 is the depth distributions characters and their impact factors and reasons.

The vertical distributions presented the character at 35 °N transect. Vertical distributions were related to the solar radiation. Unfortunately, CDOM and solar radiation were not measured in this study, we will set up these parameters in the future research to confirm the distribution driving factors.

Yes, the cited information is old, the citation in the last two sentences were deleted and some new information were cited.

"The addition of photosensitizers-natural DOM and commercial humic acid (HA) photo-catalyzed glutathione (GSH) and cysteine, and enhanced the COS formation (Flöck et al., 1997). An excited triplet state CDOM ($^3CDOM^*$) is produced by COS in the presence of ultraviolet light (Li et al., 2022)." has been added after "The high COS concentrations in the surface seawater in spring in this study may be attributed to the photochemical production reactions of $CS_2$ and COS in the euphotic zone because they are dependent on light (Flöck et al., 1997; Xie et al., 1998).".

The statements at the end of the paragraph about sulfur in the deeper sea ("It has been shown that $CS_2$ can be produced by anaerobic fermentation by bacteria and by reactions between $H_2S$ and organic matter in pore water (and anoxic basins) (Andreae, 1986). This hypothesis agreed with the results of Wakeham et al. (1987), where the concentration of $CS_2$ peaked (at about 20 nmol $L^{-1}$) near the sediment-water interface. Jørgensen and Okholm-Hansen (1985) found that the release rate of VSCs (such as $CS_2$) in surface seawater was usually 10 to 100 times lower than that in underlying sediments in a Danish estuary, indicating that release from sediments is an important source of $CS_2$.") have been deleted. The sentences of "Consistent with our $CS_2$ results, Xie et al. (1998) showed that the $CS_2$ concentrations decreased with the depth, coinciding with

solar radiation changes. Decreased photochemical reaction due to decreasing solar radiation with water depth may explain the vertical distribution of $CS_2$ (Xie et al., 1998). Similar to the results of Xie et al. (1998), the high $CS_2$ concentrations in the bottom seawater at station H15 in spring may be attributable to a sedimentary source." have been added.

Global sea-air fluxes have been added in Section 4.3. "The model of Lennartz et al. (2021) was not used to evaluate the global sea-air fluxes of DMS, OCS, $CS_2$ in this study due to a lack of parameters, i.e., the absorption coefficient of CDOM at 350 nm (a350), global radiation (converted to UV radiation), and sea surface pressure. Therefore, the global sea-air fluxes of DMS were calculated following Hulswar et al. (2022) with minor modifications. The global sea-air fluxes of OCS or $CS_2$ were evaluated by the mean sea-air fluxes of OCS or $CS_2$ multiplied by the ocean area and the time. The global sea-air fluxes of DMS, OCS, and $CS_2$ were 21.3, 2.3, and 2.0 TgS year$^{-1}$, respectively. The global sea-air flux of DMS was similar to the results of Hulswar et al. (2022) (27.1 TgS year$^{-1}$). In comparison, the global sea-air fluxes of OCS and $CS_2$ were 15.9- and 9.9-fold higher than the results of Lennartz et al. (2021). The different calculation method we used may overestimate the global sea-air fluxes of OCS and $CS_2$. The another reason may be the high sea-air fluxes of OCS or $CS_2$ in the BS and YS because marginal seas are significantly influenced by anthropogenic emissions (Watts, 2000). The sea-air fluxes of DMS, OCS, and $CS_2$ in the BS and YS were 28.2, 3.1, and 2.7 GgS year$^{-1}$, accounting for 0.10%, 2.23%, and 1.44% of global sea-air fluxes. The BS and YS comprise 0.13% of the global sea area; therefore, they contribute considerably to global sea-air fluxes."

Section 4.2 – Every possible explanation is given for the atmospheric distributions. Again, what are the findings here and the main idea of each paragraph? The discussion of the DMS values in the air need more explanation (especially related to the anthropogenic source). First of all, the atmospheric lifetime of DMS is on the order of 1 day. Therefore, 72-hour back trajectories are not appropriate. If there is a relevant anthropogenic DMS source, it needs to be stated and cited.

Response: The first paragraph discussed the results of VSCs mixing ratios in this study and previous studies. The new reference (Xu et al., 2023) has been added. The second paragraph discussed the atmospheric VSC mixing ratios are influenced by anthropogenic VSCs emissions and VSCs concentrations in seawater. The third paragraph discussed the wind direction and air masses of the back trajectories of several stations to find the sources or reasons of the high or low VSCs mixing ratios. The wind direction of air mass and the back trajectories of Miyakojima, Yokohama, and Otaru in Japan in winter of Hattori et al. (2020) have been discussed.

Explanation about the discussion of the DMS values in the air (especially related to the anthropogenic source) has been added. 72-hour back trajectories mean trajectories from 72 h to 0 h before sampling, therefore, which include 24 h~0 h (1/3 of the line near the sampling station).

The anthropogenic DMS source has been stated and cited "The wind direction is from continental Asia to the Pacific in spring. The backward trajectories of B49, B47, and

B08 showed that anthropogenic and oceanic DMS emissions accounted for the atmospheric DMS sources. The wind direction of the air mass from the back trajectories of Miyakojima, Yokohama, and Otaru in Japan in winter (January to March) observed by Hattori et al. (2020) was similar to ours in spring (March to April). Hattori et al. (2020) reported that the anthropogenic COS originated primarily from the Chinese industry and was transported by air to southern Japan. The backward trajectory of H09 showed that the wind direction was from the south of Taiwan Island in summer, and oceanic sources accounted for the atmospheric DMS.".

Supplemental material – The figures are cited out of order in the main text. Using a compromise to provide the same scale for the two plots in figure S1 might make the information more attainable. Table S2 should have references to the work providing the constants. Why are tables S3 and S4 in the supplements and not the main text? They seem like key components of the discussion.

Response: The same scales have been used in the nutrient figures in Fig. S2.

The references (De Bruyn et al., 1995; Dacey et al., 1984) for the constants in Table S2 are provided. Tables S3 and S4 in the supplements have been merged into Table 1 and moved to the main text.

Reference:

Dacey, J. W. H., Wakeham, S. G., and Howes, B. L.: Henry's law constants for dimethylsulfide in freshwater and seawater, Geophys. Res. Lett., 11, 991–994, https://doi.org/10.1029/GL011i010p00991, 1984.

De Bruyn, W. J., Swartz, E., Hu, J. H., Shorter, J. A., Davidovits, P., Worsnop, D. R., Zahniser, M. S., and Kolb, C. E.: Henry's law solubilities and Šetchenow coefficients for biogenic reduced sulfur species obtained from gas-liquid uptake measurements, J. Geophys. Res.-Atmos., 100, 7245–7251, https://doi.org/10.1029/95JD00217, 1995.

[Figure]

Figure S2. Spatial distributions of nitrate, phosphate, and silicate in the surface water of the BS and YS in spring (a)-(c) and summer (d)-(f).

**Table 1** Correlation analyses of the three VSCs and environmental factors in the BS and YS in spring and summer.

| Spring | COS (seawater) | DMS (seawater) | CS$_2$ (seawater) | COS (atmosphere) | DMS (atmosphere) | CS$_2$ (atmosphere) |
|---|---|---|---|---|---|---|
| COS (seawater) | 1 | | | | | |
| DMS (seawater) | 0.021 | 1 | | | | |
| CS$_2$ (seawater) | 0.193 | 0.281* | 1 | | | |
| COS (atmosphere) | -0.246 | -0.355 | -0.182 | 1 | | |
| DMS (atmosphere) | 0.296 | 0.04 | 0.274 | 0.117 | 1 | |
| CS$_2$ (atmosphere) | -0.201 | -0.264 | -0.213 | 0.554** | -0.013 | 1 |
| Chl $a$ | 0.132 | 0.044 | -0.095 | 0.033 | 0.179 | -0.141 |
| Temperature | 0.286* | 0.082 | 0.319** | -0.257 | 0.179 | -0.372 |
| Salinity | 0.11 | -0.009 | -0.109 | 0.24 | 0.019 | 0.236 |
| Silicate | -0.103 | -0.252* | -0.029 | 0.351 | -0.008 | 0.54 |
| Phosphate | -0.084 | -0.205 | -0.353** | 0.621 | -0.128 | 0.36 |
| Nitrate | -0.299* | -0.293* | -0.226 | 0.075 | -0.096 | 0.044 |
| DOC | -0.146 | -0.153 | -0.073 | 0.037 | -0.122 | 0.008 |
| Summer | COS (seawater) | DMS (seawater) | CS$_2$ (seawater) | COS (atmosphere) | DMS (atmosphere) | CS$_2$ (atmosphere) |
| COS (seawater) | 1 | | | | | |
| DMS (seawater) | 0.009 | 1 | | | | |
| CS$_2$ (seawater) | -0.007 | 0.424** | 1 | | | |
| COS (atmosphere) | 0.358 | 0.472 | 0.184 | 1 | | |
| DMS (atmosphere) | -0.266 | 0.404 | 0.31 | 0.451 | 1 | |
| CS$_2$ (atmosphere) | 0.452 | 0.229 | 0.424 | 0.855** | 0.251 | 1 |
| Chl $a$ | -0.059 | 0.25 | 0.274* | 0.461 | -0.294 | 0.565 |
| Temperature | 0.088 | -0.076 | -0.143 | -0.097 | -0.349 | 0.072 |
| Salinity | 0.128 | -0.172 | -0.143 | -0.12 | -0.352 | -0.044 |
| Silicate | 0.114 | 0.122 | 0.276* | 0.312 | -0.548 | 0.377 |
| Phosphate | 0.104 | -0.169 | -0.245 | -0.49 | -0.539 | -0.482 |
| Nitrate | -0.095 | 0.145 | 0.057 | -0.008 | 0.224 | -0.155 |
| DOC | 0.342* | -0.015 | 0.012 | 0.02 | 0.924 | 0.319 |

* indicates $P < 0.05$, ** indicates $P < 0.01$.

[Figure]

EditorBar Language Editing
No. 35, Tsinghua East Road, Beijing, China 100083
Email: runse@editorbar.com Phone: +86-10-5620-8614

**CERTIFICATE OF LANGUAGE EDITING**

The English writing of the following manuscript was carefully edited by a native English speaker.

**Manuscript Information**

| | |
|---|---|
| ID | LE202308120181 |
| Editing date | 2023-08-15 |
| Title | Spatial and seasonal variability in volatile organic sulfur compounds in seawater and overlying atmosphere of the Bohai and Yellow Seas |
| Corresponding author | Gui-Peng Yang |
| Language writing before editing | □ Very poor □ Poor □ Fair ■ Good □ Very good □ Excellent |
| Recommendation after language editing | □ Submitting to target journal directly
■ Submitting to target journal after minor revision
□ Re-editing required after major revision
□ Not suitable for publication |
| Overview comments | This paper required edits with regard to wording, sentence structure, punctuation, language, tense, and grammar. There were some sections where your meaning was unclear. I added comments and offered alternate wordings for these sections or asked you to clarify if I could not interpret your meaning from the context. You should check those sections carefully to ensure that I did not change your intended meaning with my edits. |

**Edited by**

**Andrea L.**
senior editor
Oregon State University
Language Editing

[Figure]

**Certificate Issued by**

**Dr. Jason Qee**

*Jason qee*

Editor in Chief
Editorbar Language Editing, Beijing, China
runse@editorbar.com www.editorbar.com

Certificate link: www.editorbar.com/order/cert/LE202308120181

---

## Author Response (AR2)

**Response to the referees and the editor**

Thank you for the comments of the editor and the referees. We have revised the manuscript according to the advices, which are marked with red words in the track-changes file.

**Response to the editor**

Dear Prof. Yang and Co-authors,
thank you for re-submitting the revised version of your manuscript. The revised manuscript has been reviewed by two reviewers. Both reviewers raised some minor but also some major concerns which need to be addressed before publication of the manuscript. Please address carefully all concerns with special consideration of the points raised by Rev. #1 (report #1).

I am looking forward to your revised manuscript.

Yours Sincerely
Hermann Bange

Response: Thank you for the comments of the editor. According to the two referees' comments, we have revised the manuscript.

**Response to the referee #1**

The revised manuscript "Spatial and seasonal variability in volatile organic sulfur compounds in seawater and overlying atmosphere of the Bohai and Yellow Seas"" Yu et al., addresses most of the reviewers' comments and concerns. Back trajectories from different sides are implement which serve as basis to partly explain differences in atmospheric mixing ratios. Furthermore, additional information about the calibrations are given. Global climatologies and databases for DMS, OCS and $CS_2$ are now included in the discussion section which serve as a comparison for the presented dataset. However, there might be an issue with the flux calculations (mentioned below) which would influence the results. Additionally, some parts of the manuscript are still unclear and need revision as stated below.

General comments
Calibration: 1.) Calibrations, shown in the supplement, reveal very different slopes and y-intercepts in between the two seasons (spring, summer). Perhaps the authors could give some information on how this is possible, as for both seasons the same standards have been used. 2.) The y-intercept for COS and $CS_2$ calibrations is very high. A high (positive) blank could be induced by e.g. contamination. $CS_2$ and COS are known to easily contaminate the sample e.g. when using non-PTFE or silicon tubing. Did the

authors check if these high "blank" values are only related to the standard used or if these blank values are also visible when measuring a "blank sample"? It would be great to hear an explanation of this issue.

Response: 1) Although the same standard was used, GC-MS exhibited different status at different seasons, including the chromatographic column status and ion source status, which may explain the different slopes and y-intercepts in spring and summer.

2) The tubes and trap used in the examination were all PTFE tubes. The high y-intercept for COS and $CS_2$ calibrations may be due to the small blank values when measuring a blank sample because of slight existence of COS and $CS_2$ in the carrier gas. There is no DMS in the carrier gas. However, the values measured in the samples were 10 fold of blank standard deviation, so the results in our research were reliable.

Henry's law constants: According to the reference in Table S2 for the calculation of the DMS Henry's constant, the given constant C1 is wrong, which would result in a different H by ~1 order of magnitude. If Dacey et al. (1984) was used, C1 should be 0.56 mol $L^{-1}$ atm$^{-1}$ instead of 0.048 mol $L^{-1}$ atm$^{-1}$. Please check if it is just a typo in the manuscript. Otherwise fluxes have to be recalculated and values including their implications for the discussion have to be revised.

Response: Yes, it is a typo in Table S2. We have changed 0.048 mol $L^{-1}$ atm$^{-1}$ to 0.56 mol $L^{-1}$ atm$^{-1}$ in Table S2.

Flux calculation discussion (ll.392): It is not clear to me why the authors recalculate global DMS, COS and CS2 fluxes. What are the presented global numbers in l. 397 based on? Did the authors use mean fluxes from their study and extrapolated them to global numbers assuming the same flux also at different locations in the world? This paragraph needs a revision.

Response: We misunderstood the meanings of the advice of the referee for the last revision. The paragraph has been deleted and we have compared our data with the database. See the 1st paragraph in sections 4.1.1, 4.2, and 4.3.

Specific comments

l.56: Please introduce $^3CDOM^*$, $^1O_2$, $H_2O_2$ and $^•OH$.

Response: $^3CDOM^*$, $^1O_2$, $H_2O_2$ and $^•OH$ have been introduced as "excited triplet states of chromophoric dissolved organic matter ($^3CDOM^*$), singlet oxygen ($^1O_2$), hydrogen peroxide ($H_2O_2$), and hydroxyl radical ($^•OH$)" in the introduction.

ll.128: According to the answers of the reviewers, the authors calculated the flux of DMS assuming the atmospheric mixing ratio to be zero. This should be mention here in the method section. Additionally, the authors should mention in the discussion section, that their calculated DMS fluxes should be seen as upper limits (due to setting the atm mix ratio to zero). Why did the authors not use atm mix ratios for the flux calculation as they are available?

Response: The $C_g$ of DMS is assumed to be zero in this study. This is based on the fact that atmospheric mixing ratio of DMS are typically several orders of magnitude lower

than concentrations in seawater (Turner et al., 1996). The sentences have been added in section 2.3.

We have mentioned that the calculated DMS fluxes should be seen as upper limits due to setting the atm mix ratio to zero in the discussion section 4.3.

ll.132: The sentence "This method has been internationally accepted" can be deleted.

Response: The sentence "This method has been internationally accepted" has been deleted.

l. 189: surface water is referred to 4m depth. Later on (e.g. l.204) surface water is referred to 3m depths. Please be consistent.

Response: The surface sampling depths for spring and summer are different, which is ~4 m and ~3 m, respectively. See the values in Fig. 4 and Fig. 5 in data at https://doi.org/10.6084/m9.figshare.14971644. In order to be consistent (3−5 m), the sentence of "Surface seawater was sampled at a depth of 3−5 m." has been added in section 2.1.

ll. 206: To my understand, a x-fold increase between A and B is defined as the ratio between A and B. Somehow, throughout the whole manuscript, it seems the authors define a x-fold increase as the ratio between (A-B) and B. Please check the correct definition or use other wording. In order to easier compare two values, I suggest to use the ratio A/B.

Response: According to the advices of the referee, x-fold increase has been changed to the ratio A/B in the revised manuscript.

ll.219: I do not understand the revised sentences about differences in atmospheric values of DMS between B47 and B49. According to the provided back trajectories for B47 and B49 (which are single runs and no ensemble or similar) I think the authors can not say that they are different. Therefore, the conclusion why B47 and B49 strongly differ in atm mix ratio is lacking.

Response: In order to see the differences, we have added the 12 h and 24 h back trajectories. The differences in the 72 h back trajectories can not be seen, while they can be observed in 12 h or 24 h back trajectories. The 12 h and 24 h back trajectories for station B47 showed that the air mass over station B47 differed slightly from that over station B49 as it traversed the land of Liaoning province. See Fig. S3. The sentence of "The air mass over station B47 differed slightly from that over station B49 as it traversed the land of Liaoning province (12 h and 24 h backward trajectories in Fig. S3)." has been added in 3.3.1.

ll.222: "high oceanic DMS concentrations at or near stations where air masses were passing through…may be the reason". I do not understand this sentence.

Response: This sentence is our speculation according to the data and the air masses pathway without confirmation. Therefore, this sentence has been removed.

ll.320: The introduction of $^3CDOM^*$ should happen when using the abbreviation for the first time.

Response: The introduction of $^3CDOM^*$ has been added when using the abbreviation for the first time in the 4th paragraph in the introduction in the revised manuscript.

ll.341: The paragraph about diurnal COS variations does not state or show any data from the actual study. It is not clear to me what exactly the authors want to highlight with this paragraph with respect to their dataset. If they did not investigate the atmospheric concentrations with respect to the light intensities or sampling time they should at least state that the sampling time could influence the measured COS atm mix ratios (besides wind speed, wind direction and oceanic concentration). Or did the authors sample the station always at the same time of the day?

Response: The paragraph about diurnal COS variations were added according to the other reviewer's suggestion. Yes, we did not evaluate the diurnal COS variations. We have simplified the discussion about diurnal variations, and the sentences "The maximum COS concentration occurred 3 h after the maximal global radiation intensity (COS: 15:00; global radiation intensity: 12:00) due to the balance between COS production and removal (Xu et al., 2001)." have been deleted, and pointed that "Therefore, the sampling time can influence the measured COS concentrations in the seawater.".

Figure 6: Please increase the resolution of the figure (higher quality). I was hard to see small dots (low concentrations) on the map.

Response: The small dots for low concentrations in Figure 6 have been enlarged and increased the resolution of the figure. We can provide the TIF formation for the figure to increase the resolution, if needed.

**Response to the referee #2**

The authors addressed most of my comments and a good number of the other reviewer's. However, I still have some issues with the manuscript. My specific comments are below (line numbers refer to track changes manuscript):

Introduction - this whole section is still a bit unclear. There is a whole paragraph dedicated to COS (2nd) and one for DMS (3rd), but $CS_2$ does not have the same. In the 4th paragraph (and on), all the compounds are mixed together, which makes it (them) somewhat confusing. There is no introduction sentence in the paragraphs that helps to set the main ideas.

Response: The sentences in the 4th paragraph have changed or added as "The photochemical reaction of DOM generates excited triplet states of chromophoric dissolved organic matter ($^3CDOM^*$), singlet oxygen ($^1O_2$), hydrogen peroxide ($H_2O_2$), and hydroxyl radical ($^•OH$). These reactive species subsequently interact with DMS, resulting in the production of $CS_2$ (Modiri Gharehveran and Shah, 2021). The oxidation

reaction involving the OH radicals and $CS_2$ is a substantial contributor to the generation of $SO_2$, which subsequently leads to the production of acid rain (Logan et al., 1979).". The sentence "Production and loss processes of COS, DMS, and $CS_2$ have been documented by many researchers in the following manners." has been added in the 1st paragraph to help to set the main ideas of the 2-4 paragraphs.

Lines 43-44 - This sentence is repetitive (following the sentence before). Either take out this idea from the previous sentence or remove this sentence.
Response: The referee is right and the sentence is repetitive. The sentence of "COS production rates increase with increasing nitrate concentration (Li et al., 2022)." has been removed.

Lines 56-57 - The English is not clear here.
Response: The sentence "$^3CDOM^*$, $^1O_2$, $H_2O_2$, and $^•OH$ produced by the photochemical reaction of DOM react with DMS and produce COS and $CS_2$ (Modiri Gharehveran and Shah, 2021)." has been changed into "The photochemical reaction of DOM generates excited triplet states of chromophoric dissolved organic matter ($^3CDOM^*$), singlet oxygen ($^1O_2$), hydrogen peroxide ($H_2O_2$), and hydroxyl radical ($^•OH$). These reactive species subsequently interact with DMS, resulting in the production of $CS_2$ (Modiri Gharehveran and Shah, 2021).".

Lines 66-68 - This is repetitive in comparison to an earlier paragraph that talks about sources and sinks of COS. I am not sure why it is repeated here again. I think it should be removed as it doesn't add any additional information. In this paragraph, please just focus on what your study does and why it is relevant/important.
Response: The sentences of "Yu et al. (2022) investigated the distributions of COS, DMS, and $CS_2$ and sea-to-air flux in the Changjiang Estuary and the adjacent East China Sea, demonstrating that oceanic VSCs (COS, DMS, and $CS_2$) are sources of atmospheric VSCs. In contrast, Zhu et al. (2019) showed that the ocean was a COS sink." have been removed.

Lines 250-251 (261-262, 385) - I know the other reviewer asked for stats in relation to how the wind is a main controlling factor for the flux, but I think this is unnecessary. You use wind to compute the flux, often with a quadratic dependence. Therefore, if the wind is not correlated to the flux, there would be a problem. I think this should be removed - wind is an obvious controlling factor. The statement on line 249 -250 is appropriate.
Response: Yes. We use wind to compute the flux and wind is an obvious controlling factor. Therefore, lines 250-251 (261-262, 385) have been removed.

Section 4.1.1 - you mention pollution as a factor for $CS_2$ (also for DMS), in the context of seawater concentrations, but is there any evidence of invasion? You state that the ocean is a major source to the atmosphere for all gases, which means to me that the oceanic distribution controls what is in the atmosphere (not the other way around).

Response: No, there is no direct evidence of invasion, and this is a speculation. The sentence of "For example, rayon production is the main source of anthropogenic $CS_2$ (Campbell et al., 2015) in the northern cities of the BS." in section 4.1.1 has been removed.

Thanks a lot for the advice. The atmospheric sources are not only oceanic distribution. Therefore, the sentences in the abstract and conclusion have changed "major" into "important".

Lines 316-321 - There is much focus on sources of the gases, but could slower or less loss also be responsible here?

Response: The sentences of "In addition, the loss processed include exhalation, downward mixing, and hydrolysis. Among these processes, hydrolysis is the main sink (Xu et al., 2001). Slow hydrolysis rate may be another reason to explain the high COS concentrations in the surface seawater." have been added.

Section 4.3 – I am a bit confused by the discussion and revision of this section. The idea was not necessarily to use the method of Lennartz et al. (2021) to compute the fluxes for COS, but to compare to the database. How do your calculations match other comparable areas (i.e., marginal seas). Then why is there a therefore on line 394 about how DMS was computed? Also, why are global fluxes extrapolated? I think these findings simply need to be compared to others to give context. Finally, again, I do not understand the argument about anthropogenic emissions. Are the anthropogenic emissions from runoff or so, so that they go directly into the water?

Response: We misunderstood the meanings of the advice of the referee for the last revision. The paragraph has been deleted and we have compared our data with the database. See the 1st paragraph in sections 4.1.1, 4.2, and 4.3.

No, the anthropogenic emissions were not meant from runoff. They mean coal combustion, industrial production, et al., which contribute to the VSCs mixing ratios in the atmosphere. The anthropogenic emissions discussion has been removed because of the revision in this paragraph.

A note on back trajectories – I want to make sure I am clear here…a back trajectory of 72 hours is not useful for DMS. It is fine if you say that only a few of the points are useful (the ones that represent the last 24 hours), however, you should state where those are. Is each dot 12 hours? If so, the trajectory, for example, over B49 does not pass over land within that time period. Therefore, land sources are not a good explanation for DMS.

Response: The 12 h and 24 h back trajectories for DMS has been added in Fig. S3. "The air mass over station B47 differed slightly from that over station B49 as it traversed the land of Liaoning province (12 h and 24 h backward trajectories in Fig. S3)" (added in section 3.3.1), where DMS may loss when passed over the land." See Fig. S3.

A statement of the atmospheric lifetimes of the compounds of interest should be made and compared with the trajectories. For COS (long-lived), the trajectories are

meaningful. Land sources of COS should be stated clearly. What about $CS_2$? The lifetime is about 1 week, so the land sources are important for it too. You say that the spring back trajectories show anthropogenic influences, but how? Do you see reduced fluxes of COS and $CS_2$ when the back trajectories come from a certain area? Is there visible trend? All but one of the back trajectories show land influence. Was H09 in summer anomalous? I think the discussion of this information is not deep or always meaningful.

Response: A statement of the atmospheric lifetimes of the compounds has been added in section 4.2 to show the meaning of the trajectories.

Spring back trajectories show anthropogenic influences, including coal combustion, industrial production, et al. It is a speculation, because the spring back trajectories traversed the land.

No, we did not see. Not all back trajectories maps were drawn, and we only drew them for some stations.

No, H09 in summer was not anomalous, it is only an example for oceanic source.